# How hard are computer vision datasets? Calibrating dataset difficulty to viewing time

**David Mayo**\*
CSAIL & CBMM
MIT

**Jesse Cummings**\*
CSAIL & CBMM
MIT

**Xinyu Lin**\*
CSAIL & CBMM
MIT

**Dan Gutfreund**
MIT-IBM Watson AI Lab
IBM

**Boris Katz**
CSAIL & CBMM
MIT

**Andrei Barbu**
CSAIL & CBMM
MIT

## Abstract

Humans outperform object recognizers despite the fact that models perform well on current datasets, including those explicitly designed to challenge machines with debiased images or distribution shift. This problem persists, in part, because we have no guidance on the absolute difficulty of an image or dataset making it hard to objectively assess progress toward human-level performance, to cover the range of human abilities, and to increase the challenge posed by a dataset. We develop a dataset difficulty metric MVT, Minimum Viewing Time, that addresses these three problems. Subjects view an image that flashes on screen and then classify the object in the image. Images that require brief flashes to recognize are easy, those which require seconds of viewing are hard. We compute the ImageNet and ObjectNet image difficulty distribution, which we find significantly undersamples hard images. Nearly 90% of current benchmark performance is derived from images that are easy for humans. Rather than hoping that we will make harder datasets, we can for the first time objectively guide dataset difficulty during development. We can also subset recognition performance as a function of difficulty: model performance drops precipitously while human performance remains stable. Difficulty provides a new lens through which to view model performance, one which uncovers new scaling laws: vision-language models stand out as being the most robust and human-like while all other techniques scale poorly. We release tools to automatically compute MVT, along with image sets which are tagged by difficulty. Objective image difficulty has practical applications – one can measure how hard a test set is before deploying a real-world system – and scientific applications such as discovering the neural correlates of image difficulty and enabling new object recognition techniques that eliminate the benchmark-vs-real-world performance gap.

## 1 Introduction

Numerous efforts exist to build better evaluations for object recognizers. Broadly, these fall into four categories. Those that probe distribution shift, like ImageNetV2 [1]. Those that add scale like OpenImages [2]. Those that explicitly attempt to make images more difficult for models by adversarially selecting them, like ImageNet-A [3] or adding artificial corruptions, like ImageNet-C [4]. And those that attempt to explicitly control for biases like ObjectNet [5]. These are responses to the fact that performance on standard benchmarks does not translate well to real-world conditions;

---

\*Equal contribution. Website `https://objectnet.dev/mvt` Corresponding author `dmayo2@mit.edu`

37th Conference on Neural Information Processing Systems (NeurIPS 2023) Track on Datasets and Benchmarks.

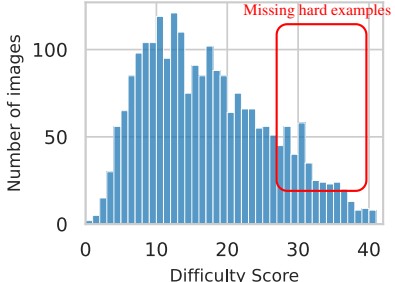 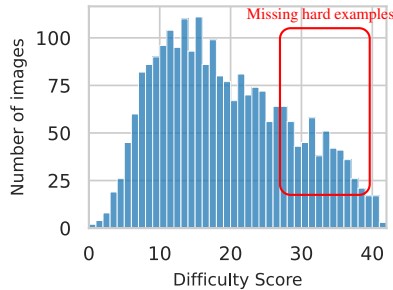

(a) ImageNet image difficulty distribution    (b) ObjectNet image difficulty distribution

Figure 1: ImageNet and ObjectNet image difficulty distributions. Difficulty here is defined as how many participants failed to recognize a given image across viewing times. The difficulty of both datasets is roughly the same, significantly under-sampling hard images. Slightly under 90% of performance metrics are derived from easy images. Today, we largely as a community only test what is easiest for the human visual system.

90% accuracy for one class in ImageNet does not mean that the detector will achieve 90% accuracy for that class in one's home or on frames of a movie. In all four cases, these efforts have no objective guide, no metric that evaluates how far they have progressed towards enabling models to generalize.

We set out to measure an orthogonal quantity – how difficult images in these datasets are for humans. Distribution shift and bias control won't on their own address this problem if datasets are overwhelmingly easy compared to what humans are capable of recognizing. And while scale helps, if datasets are heavily skewed toward images that are easy for humans, the statistics of performance on such datasets may hide the real underlying performance trends of models on harder images.

An objective metric by which to measure the difficulty of computer vision datasets has several advantages. First, we can determine if there are gaps in our datasets; perhaps certain difficulties are systematically undersampled. We find that this is the case: hard images are essentially missing. Moreover, merely aiming for distribution shift, even by changing how the dataset is gathered, doesn't meaningfully change the difficulty distribution; ObjectNet and ImageNet were gathered from different sources (captured by Mechanical Turk vs the web), with different goals and additional controls for ObjectNet, yet their difficulty distributions are remarkably similar. Second, we can evaluate model scaling as a function of difficulty. We find that most model families scale poorly in terms of out-of-distribution robustness, generalizing well on the easy images but hardly improving on the hard images, with the exception of CLIP [6]. Third, it provides a new kind of metric for biological plausibility, orthogonal to raw performance, error distribution, or how well networks predict neural activity. If a network is to be a model of the human visual system, not just an engineering model, then some quantity computed from that network should explain the observed difficulty scores. About half of the variance in the difficulty results is accounted for by a combination of c-score [7], prediction depth [8], and adversarial robustness [9]. Fourth, tools that measure difficulty could be incorporated into dataset collection and into how we report datasets and the overall progress of our community. In the long term, we intend to establish a dashboard giving a perspective on object recognition from the point of view of difficulty.

To build this difficulty metric, we choose as a proxy the minimum viewing time (followed by a backward mask) that a human viewer requires before being able to recognize the object in an image. Earlier readout is likely an indication that fewer mental resources were needed to recognize the image. After viewing the image, subjects have unlimited time to respond to a 1-out-of-50 forced choice task where they must identify the object class in the image that was shown. This metric is related to object solution time (OST) [10] explored in the neuroscience literature. We are of course not the first to carry out such viewing time experiments [11], but we do so at scale and with images from modern datasets. Further, we turn these results into a difficulty metric with practical applications, predict this difficulty metric from quantities computed from current networks, and show the scaling of current models. We hope that in the future, benchmarks will regularly report their difficulty distribution (they can do so for only a few hundred dollars with the tools we provide) and that collections of benchmarks will seek out datasets based on their difficulty distribution. Additional attention may need to be paid while collecting datasets to not eliminate hard examples; any quick consensus-based process with multiple annotators is likely to be heavily biased against including hard examples. Practically, when collecting datasets for domains where the cost of errors is high (the medical domain, autonomous vehicles, etc.),

being mindful of the difficulty distribution and actively shaping it to fill out harder images may be critical to building confidence in the resulting models.

Our contributions are:

1. A dataset of 200,382 human object recognition judgments as a function of viewing time for 4,771 images from ImageNet and ObjectNet
2. The first objective image difficulty metric, minimum viewing time, MVT
3. The difficulty distribution for ImageNet and ObjectNet
4. An evaluation of model performance as a function of image difficulty and a new scaling law
5. A new metric for validating the biological plausibility of models, predicting image difficulty
6. A new subset of images from ObjectNet and ImageNet sorted by difficulty for use in neuroscientific and behavioral experiments.

## 2 Related work

**Image difficulty** Image difficulty has previously been investigated primarily from the perspective of models. Jiang and Zhang et al. [7] constructed a new metric for image difficulty, c-score, which is well approximated by its learning speed proxy; more difficult images are classified correctly later in the course of training. Agarwal et al. [12] introduced variance of gradients (VoG), a method similar to the c-score learning speed proxy, but using the variance in the gradient updates. Baldock et al. [8] identify the first layer at which a model produces the same classification as its final output. These methods find ObjectNet significantly more difficult than ImageNet, unlike our human presentation time metric as shown in fig. 1 because ObjectNet is harder for models; they are fundamentally not objective metrics of difficulty but they depend on the training set, architecture, optimizer, etc. of a specific model. Other work [13, 14] has found that image difficulty estimates can improve task performance or inform compute trade-offs. Such approaches are promising and could likely benefit from more objective, model-free difficulty measures like ours. Meding et al. [15] investigated image classification accuracy as a function of model type, finding that many images are solved by all models, while some are never solved. Humans predicted which images were easy vs. hard. This is an overt binary judgment that requires introspection, while our work provides a graded difficulty metric which is implicit to the operation of the visual system.

**Dataset design** As a field, we have measured our progress by models' performance on tests created by splitting a random subset of images from large-scale image datasets. Most datasets were created by web-scraping images and labeling them according to consensus of human annotatators [16]. More challenging test sets were developed by recreating existing test sets from new data (ImageNetV2 [1]), constructing adversarial test sets (ImageNet-A [3]), adding corruptions to existing test sets, (ImageNet-C [4]), and eliminating spurious correlations (ObjectNet [5]). Others have improved labeling errors in existing datasets to demonstrate, much like our own work, that model performance gains are misleading and that we should not declare victory too soon [17, 18]. Our image difficulty metric can aid all of these datasets by calibrating their difficulties and providing a roadmap for how to collect better datasets.

**OOD model performance** Prior work studying object recognition models performance has found a linear trend in performance improvement on ImageNet and other OOD datasets [1, 5, 19]. This linear trend can be beaten slightly when massively scaling up the quantity of training data used. Recently, the multimodal models CLIP [6] and LiT [20] have demonstrated a break from this linear trend, greatly increasing out of distribution generalization performance by using multimodal learning.

**Human Judgments** Collecting human behavioral data has long been integral to computer vision research. Human psychophysics has been used to improve model robustness [21] and representational power [22], to motivate evaluation metrics [23–27] and to develop challenging datasets with rich annotations [28, 29]. Modeling human behavioral data is at times an end in itself [22, 30–33]. These endeavors often find that while models have made progress toward parity with humans, they continue to lag behind in many important dimensions [34]. Especially relevant to our work is ImageNet-X, a labeling of ImageNet with human judgment annotations across 16 dimensions (pose, lighting, etc.) designed to aid in discovering and explaining failure modes of models with higher resolution. While these dimensions are a great contribution, they fail to explain the more complex phenomenon of human recognition difficulty.

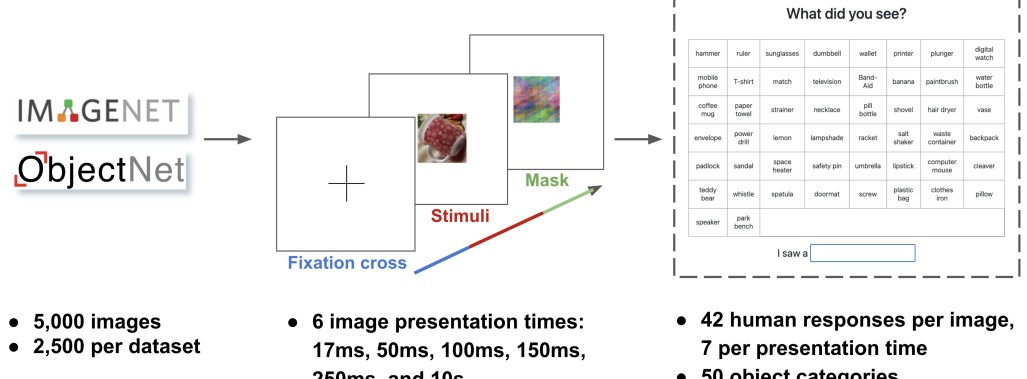

- 5,000 images
- 2,500 per dataset

- 6 image presentation times: 17ms, 50ms, 100ms, 150ms, 250ms, and 10s

- 42 human responses per image, 7 per presentation time
- 50 object categories

Figure 2: Overview of the experiment. (left) 50 images from 50 object classes were randomly selected from both ImageNet and ObjectNet to total 5,000 images; of which we analyze 4,771. Images were cropped in a square around the object of interest and then shown to human subjects on Amazon Mechanical Turk and in a controlled laboratory setting. (middle) Participants first saw a fixation cross for 500ms, then the image for either 17ms, 50ms, 100ms, 150ms, 250ms, or 10s, followed by a mask. After each image, subjects were given a 1-out-of-50 forced-choice task to identify the correct object class. (right) Each image was seen by 42 subjects, seven for each of the six image durations. No subjects saw the same image twice.

**Limited viewing time** Geirhos et al. [35] studied human and machine performance under limited presentation time with images altered by corruptions, finding that humans are more robust than machines. Rajalingham and Isaa et al. [11] presented images at 100ms finding that models are unable to match image-level behavioral patterns of primates. While our experiments agree with these results, they provide new capabilities: measuring the difficulty of entire datasets, calibrating dataset difficulty to human abilities, discovering new scaling laws, etc. All of these are derived from revealing a new understanding of difficulty as a function of image viewing time. Geirhos et al. [36] investigated the humans-machine gap as a function of architecture, objectives, and dataset sizes using a 16-way classification experiment with 200ms presentations. In this work, we investigate an orthogonal quantity, the human-machine gap as a function of viewing time, and derive a new metric from it. Our results agree with this prior work and expand upon it.

## 3 Experiment

We performed an experiment with human subjects on Mechanical Turk and in the lab in order to determine the minimum amount of viewing time required before subjects could identify an object present in an image. See fig. 2 for details. In what follows, we describe the stimuli, procedures, and validation of the online experiment with in-lab experiments.

**Stimuli** We selected 2,500 images from the ImageNet validation set (which contains 50 images per class; we selected all 50 for 50 classes) and 2,500 images from the ObjectNet dataset (ObjectNet is only a test set). These images were evenly distributed among 50 object classes shared between ObjectNet and ImageNet. Since ObjectNet was gathered by participants taking pictures of objects in their homes, all of the object classes are household objects (see right panel on fig. 2 for a full list of the object classes) and all participants were likely familiar with all object classes used. The 50 classes were picked to minimize similarity between classes.

At short presentation times, subjects do not have time to fixate on multiple locations. Off-center objects would be recognized by peripheral vision. To eliminate this effect, we cropped all images around the target object using the ImageNet validation bounding boxes and new bounding box annotations for ObjectNet. In each case, we produced a square 224 by 224 image, padding with black if needed. Details about cropping can be found in the appendix. Note that this eliminates clutter and focuses only on the appearance of the object; we discuss this limitation in the conclusion.

**Procedures** The MIT IRB oversaw the experiment. Subjects were consented and provided a warning about image flashing. Only subjects with normal corrected vision were included. Written instructions and a sample video of the experiment were provided. Subjects then read through a list of

the 50 object categories. The Mechanical Turk experiments had calibration steps to determine the screen size (utilizing credit cards since they have a standardized size) and distance from the screen using a blind spot test [37]. Images were shown at 8 degrees of visual angle. No private information was recorded and no offensive images were shown. Details about setup and calibration are available in the appendix.

For an overview of the procedures, see the middle panel of fig. 2. Each trial had a randomly selected presentation time, either 17 ms, 50 ms, 100ms, 150 ms, 250ms, or 10 seconds; timings were selected to account for a participant's use of a 60Hz screen and to probe fine-grained distinctions between image difficulty most apparent at short timings. We selected timings shorter than the typical human pre-saccade latency ( 250ms) so participants had time for only one fixation – with the exception of our 10s control accuracy timing. In each trial, a fixation cross was shown for 500 ms followed by the image, for one of the six timings. Immediately after the image, a phase gradient backward mask [38] was shown for 500 ms to disrupt further processing of the image. Details of the mask generation are described in the appendix.

Participants were then presented with a grid of the names of 50 object categories and were asked to click on the class of the object that they saw. As this was a forced-choice experiment, subjects were told to select their best guess if they were unsure. The order of the object classes in the grid was randomized at each trial. Subjects were given an unlimited amount of time to make a choice although this time was recorded and appears to be anti-correlated with performance (quick decisions were likely to be more accurate than slow ones).

Experiments were counterbalanced. In total, each image was viewed by the same number of subjects at the same number of viewing times. Any one subject did not see the same image twice at any duration. Each participant viewed 50 images, one from each class, 25 from ImageNet and 25 ObjectNet, evenly spread across the six viewing times. The order of images, viewing times, and classes were randomized for each participant. Participants were allowed to complete several experiments; for those who did, we ensured that they saw disjoint images each time. In total, 2,647 workers took part in the experiment. Subjects on Mechanical Turk were compensated at over $10 per hour. In total, we collected 210,000 trials, after which we discovered that a small number of images (229 images) were either incorrectly annotated, incorrectly cropped, or workers circumvented our automated means for ensuring that no two workers saw the same image twice. This resulted in 200,382 trials for 4,771 images (42 presentations of each image, each of 7 subjects seeing each image at one of six timings).

**In-lab validation**   Great variation exists between monitors and browsers, with some showing the same images for nearly twice as long at short presentation times. Subjects as well can vary in their attention or alertness. To account for this, we validated the results with experiments in the lab. These followed the same procedures described above with a 144Hz gaming monitor (27 inch LG-27GN950-B; 1ms GTG response time), on a machine with an NVIDIA 3060Ti, using Chrome 102. We recorded the screen at 200fps and found that the presentation times accurately reflected the time periods during which this monitor showed each image.

We selected 200 images from the 5,000 used in the previous experiment, evenly distributed across the 50 classes and the two datasets (ImageNet and ObjectNet). Each participant saw all 200 images, at random presentation times in a random order; they never saw the same image twice, and could only participate in the experiment once. Subjects were instructed to stay at a fixed distance from the screen. An experimenter was present throughout to ensure that subjects were not distracted. Lighting was kept constant by closing the blinds. In total, 12 subjects (6 male, 6 female) participated in this experiment, and as can be seen in the next section, there was widespread agreement between the in-lab and Mechanical Turk experiments. It took subjects just short of an hour to complete the 200 images. Subjects were compensated $20 for their participation.

## 4   Results

We considered an image as recognized at some viewing time when half of the participants could classify it. Chance on the 1-out-of-50 task is 2%; even a single correct response is an indication that something could be recognized. MVT is the minimum duration at which an image is recognized (MVT of 100ms means the image is not recognized by the majority of participants at $< 100$ms and correctly recognized by the majority of participants at $\geq 100$ms). Typical images by MVT are shown

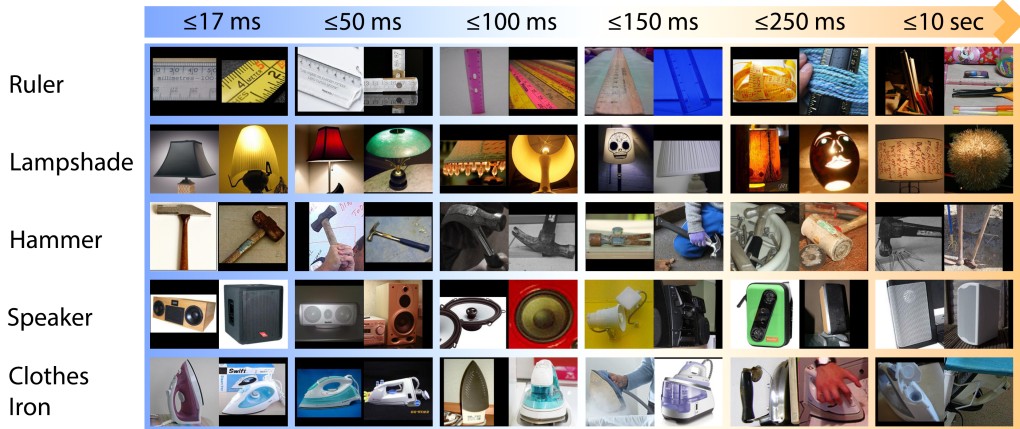

Figure 3: Images as a function of difficulty, using the minimum viewing time (MVT) before they were reliably recognized (columns), i.e., when more than half of subjects were correct. Harder images to the right are more atypical, have more difficult lighting, more occlusions, and are sometimes more blurry. Easier images to the left are more prototypical instances of their respective object class. Additional examples are available in the appendix and online. Our experiments indicate that many datasets oversample easy images.

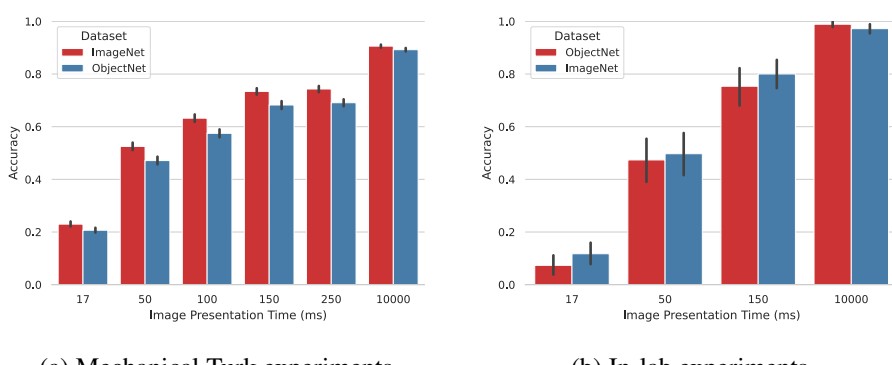

(a) Mechanical Turk experiments        (b) In-lab experiments

Figure 4: Accuracy as a function of presentation time: on the left are results for Mechanical Turk experiments and on the right are results for in-lab experiments. The same images (5,000 online, 200 in lab) were presented at 4 timings. Results for both conditions were similar, although in-lab experiments achieved nearly 100% accuracy with 10 second viewing times, while halving the performance at 17ms compared to MTurk. The accuracy improvement at long times is likely because of the more controlled conditions with fewer distractors in lab. The lower accuracies at short timings are likely due to issues with displaying images at short timings: some monitors are very slow and can fade the image in and out effectively displaying it for twice the intended 17ms.

in fig. 3. Images that are quickly recognized by humans—easy images—are most similar to those seen in datasets, while harder images include occlusion or difficult lighting.

An overview of accuracy as a function of viewing time online and in the lab is shown in fig. 4. Both experiments broadly agree with one another. In-lab experiments have higher variances due to having 20x fewer images and half as many subjects per image. In lab, the performance on short timings was significantly worse, half of that seen online. We believe this is largely due to slow monitors which display the image for significantly longer than 17ms, roughly twice as long. Recording screens with high-speed cameras supports this hypothesis. At the high end, subjects in lab were nearly 100% accurate, 10% higher than online. We believe this likely has to do with how distracted subjects were. In what follows, we focus on the Mechanical Turk experiments due to their scale.

Human performance drops off steeply when difficult images are shown for shorter viewing times; see fig. 5. This makes the experiment sensitive to even minor variations in difficulty.

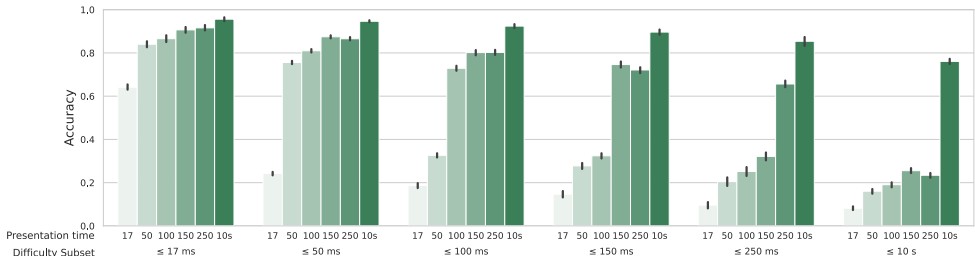

Figure 5: Human accuracy as a function of MVT image difficulty subset. Colors denote presentation time, shown below each bar, collected by minimum viewing time, shown at the bottom. Accuracy drops off steeply when hard images are displayed at short time intervals. These results are derived from Mechanical Turk experiments; in-lab experiments have exactly the same trend but higher absolute accuracies for correctly recognized images.

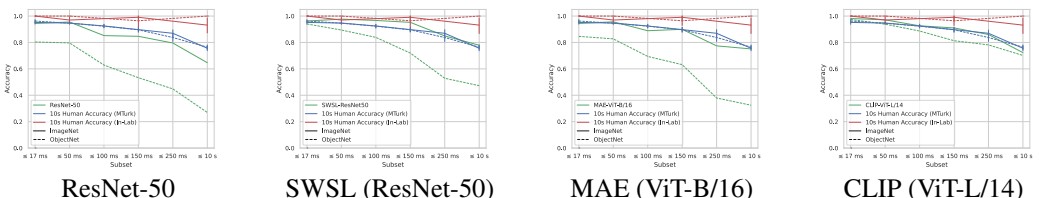

| ResNet-50 | SWSL (ResNet-50) | MAE (ViT-B/16) | CLIP (ViT-L/14) |

Figure 6: Accuracy of models on ImageNet (solid) and ObjectNet (dashed) as a function of image difficulty; many more models are reported in the appendix. Note how in in-lab experiments, human accuracy barely declines as a function of difficulty. while model performance drops off significantly. Subsetting difficult images provides much more overhead for model improvement. The in-lab human accuracy is computed over the subset of images shown in the in-lab experiments.

## 4.1 How hard are today's object recognition datasets?

The minimum viewing time required for reliable recognition is a proxy for image difficulty. Datasets today are not gathered to control for difficulty and, indeed, when plotting the difficulty of images in ImageNet and ObjectNet, we find that the difficulty curve for these datasets is highly skewed; see fig. 1. Rather than plotting six bins, one for each viewing time, we plot a more fine-grained quantity: the total number of incorrect responses out of the 42 presentations of each image (7 participants at 6 timings). Images with few incorrect responses are easy: all participants at all timings could recognize them, even at short timings. Images with many incorrect responses are hard: few participants could recognize the images at only a few longer timings.

Our results indicate that hard images are vastly underrepresented in today's datasets. The difficulty distribution of ImageNet and ObjectNet are very similar to one another. This is despite the fact that the latter was collected in a manner intentionally intended to highlight more diverse and less biased images. Merely collecting more images does not appear to help which has strong implications for how this difficulty distribution looks for other datasets that are not measured here. Instead, we must develop new tools to guide dataset collection toward more difficult exemplars; we describe such a tool and workflow below.

## 4.2 What we can learn about models from hard images

Machine accuracy varies as a function of the minimum viewing time required to recognize the object in an image. Most models—but not all—see a significant performance dropoff between the easy images and the hard images, see fig. 6. See the appendix for extensive results for dozens of models broken down by image difficulty. Note that this understates human performance as it shows the Mechanical Turk results. In-lab, even for the hardest images, humans have nearly perfect performance.

These results also show that the gap between ImageNet and ObjectNet performance increases as image difficulty increases. Likely, many more phenomena are much more acute for harder rather

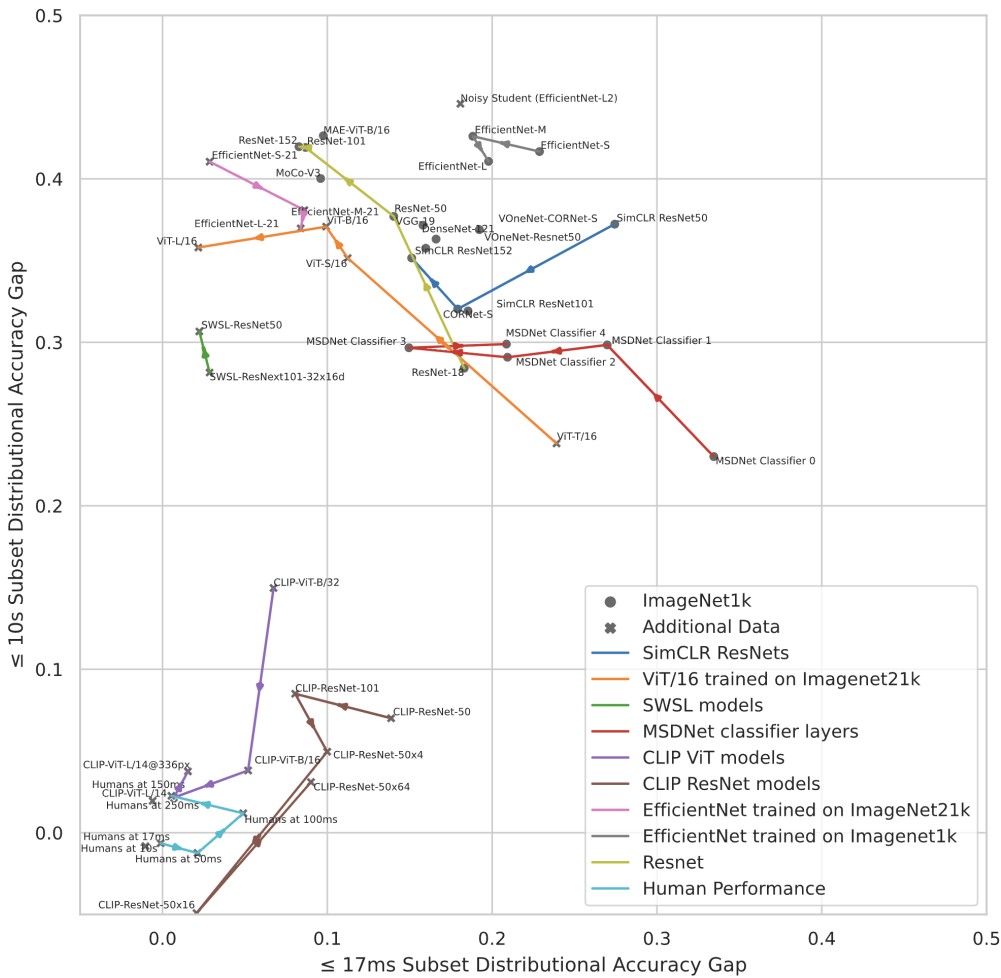

Figure 7: Robustness as a function of image difficulty comparing the ImageNet-ObjectNet performance gap on easy images (horizontal) and hard images (vertical) as determined by the MTurk MVT experiment. Each point is a model. Arrows connect model families, pointing from small to large variants. Horizontal model families are undesirable: they scale by improving on easy images. Diagonal model families are desirable: they scale by improving on both hard and easy images. Humans show no gap between ObjectNet and ImageNet. All model families are horizontal, regardless of dataset, supervision, training regime, or architecture. Only CLIP models stand out by scaling in a more human-like way. This scaling law has not been observed before and the reasons behind it are unknown.

than easier images just as distribution shift. Datasets intended to challenge object recognizers would benefit from sampling hard images, rather than vastly oversampling easy images as they do today.

Image difficulty can tease apart differences between recognition models that would otherwise be lost because of the skewed underlying difficulty distributions in current datasets. In fig. 7, we plot numerous object recognition models contrasting their performance on the easiest and hardest images. Rather than computing the absolute performance of models, which would naturally favor larger models with larger training sets, we measure the gap in performance between ImageNet and ObjectNet. A more robust detector is one that has a smaller gap, even if its performance is lower, as scaling models up (both in parameter size and training set size) is well understood. Models that are part of the same family are connected by arrows starting with the smaller variants pointing to the larger ones.

Humans hover around zero; they are robust with respect to the distribution shift between ObjectNet and ImageNet. Some model families are roughly horizontal, like SimCLR, or even have a negative slope. This shows that as they scale, their performance is increasing on the easy images, but not on the harder images, or in the case of negative slopes, it is widening the gap on harder images. CLIP stands out, and most closely approaches the human results.

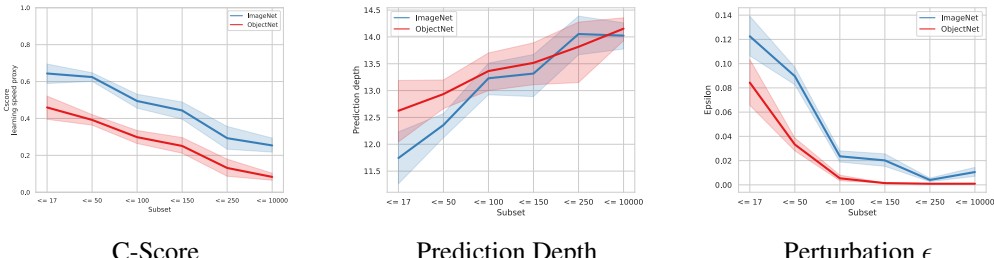

| C-Score | Prediction Depth | Perturbation $\epsilon$ |

Figure 8: The correlation between three metrics and image difficulty. On the x axis images subset by the minimum amount of viewing time, in ms, required for the majority of participants before they are recognized. All three metrics are correlated with difficulty. Hard images are learned later in model training, predicted by later layers, and need much smaller perturbations to attack.

Extrapolating the performance of models on such a graph shows that many current model families are likely to stop or radically slow down performance improvements on hard images, a much less optimistic story than if one merely considers aggregate performance. At the same time, CLIP shows promise, although it too appears to eventually begin to move away from being robust to distribution shifts as model size increases. Note that these experiments use the largest CLIP models available to the public; other large models described in the CLIP publication may have even more optimistic results. SimCLR also stands out from the non-CLIP models, demonstrating a mild productive scaling trend although it is not clear whether this trend continues with larger models.

Four weeks of compute on two machines with 8 TITAN RTX were used to generate the results in this paper. We release our data under the Creative Commons BY-SA license.

## 4.3 Explaining image difficulty and testing how similar models are to humans

If models are to not just perform well, but to also process images in ways that are similar to how the human visual system does, then they should contain a proxy for difficulty. Most models today are not recurrent, so no direct analog exists to viewing time (although, even if they were recurrent, it is unclear if directly taking the number of iterations of the model is the correct analog of viewing time). Minimum viewing time judgments provide an independent and complementary metric by which to evaluate how similar models are to human brains, in addition to their behavior, error pattern, and ability to explain neural recordings.

We investigate three quantities computed from models that could be used to explain the difficulty judgments: c-score [7], prediction depth [8], and adversarial robustness [9]. C-score is a consistency score computed while the network is training; we use the learning speed proxy to compute it. Prediction depth measures the earliest layer at which a network produces its final prediction. And adversarial robustness refers to the perturbation, $\epsilon$, needed to fool the network on a given example. See fig. 8 for an overview of the correlation between these metrics and difficulty. The same network with a ResNet-50 architecture trained on ImageNet was used for all experiments presented here. Note that $\epsilon$ is only computed over the correctly classified images.

This analysis reveals that images that require more viewing time for humans are harder for networks in several ways. They are learned much later in the training process. They are predicted by later layers in the network. And they require much smaller perturbation to create adversarial examples. This underscores that human minimum viewing time has many practical consequences for how networks process images. At the same time, these metrics alone do not explain all of the difficult judgments. Logistic regression including all three metrics classifies images into the three difficulty bins (short: (17, 50), medium: (100, 150, 250), and long (10s)) with 47.7% accuracy. Above chance, but leaving plenty of room to do more. New metrics for understanding how networks process images could be validated against this data.

We also explored more intuitive explanations of image difficulty by leveraging ImageNet-X, the human-annotated dimensions of the ImageNet validation set [29]. However, we found no correlation between these annotations and MVT. Though ImageNet-X is useful for analyzing model failure modes, it fails to meaningfully explain human psychophysics table 5. Low-MVT and high-MVT images are qualitatively distinct fig. 3, and, empirically, humans are able to distinguish between hard

and easy images [39], but a more specific explanation eludes us at this time. Just like it is hard to pinpoint in language what exactly is difficult about computer vision in general, it may also prove challenging to establish semantic explanations of image difficulty.

One high level qualitative trend that we can observe from our data is that images which are easy for humans are more prototypical while difficult images are atypical in innumerable different ways. We do intend to investigate this topic further in future work.

## 5    Conclusion, limitations, and future work

Rather than focusing on scaling, distribution shift, or control for biases alone, we should also focus explicitly on dataset difficulty. Today's datasets skew toward being too easy by undersampling hard images. ObjectNet was designed for distribution shift and bias control and was not collected from the web, yet its distribution of image difficulties is remarkably similar to that of ImageNet. By focusing on ways to measure dataset difficulty as datasets are collected, we can better calibrate the entire community, and create the resources needed to push object recognition forward. In addition to just creating better datasets, understanding performance as a function of difficulty reveals radically different scaling curves for different models and approaches. It can also provide subsets of images that highlight different types of processing for neuroscientific or behavioral experiments.

Our approach to measuring the difficulty of object recognition tasks focuses specifically on object instances, rather than other factors that also add complexity such as saliency and clutter. By cropping images around the object, we make figure-ground segmentation far easier and separate out the issue of the difficulty of this specific instance of an object as opposed to the difficulty of finding this object within a cluttered background.

The particular notion of image difficulty derived from this experiment is just one metric of difficulty. As we describe in the introduction, other metrics exist which can be computed automatically given models that recognize objects. But those metrics rely on models and therefore change as models change. The metric we present here is both absolute – it is calibrated to what humans can achieve – and model-agnostic. Models change over time, far faster than datasets do; there is a real danger when designing datasets that their difficulty will be tuned to specific models, which can be avoided by using measures related to humans.

Having performed extensive experiments to validate this approach, measuring the difficulty of any dataset is now easy and cheap. One can sample a few hundred images and run an experiment on Mechanical Turk. This only costs on the order of hundreds of dollars per dataset and can be carried out quickly. The more critical the dataset, and the cost of object recognition failures, the more important it is to do so. We provide a toolkit with code for creating experiment stimuli, hosting the experiment, posting tasks to MTurk, and collecting and analyzing data at `https://github.com/dmayo/MVT-difficulty`. All the user needs to do is provide the images.

Collecting datasets without considering image difficulty results in datasets that are skewed toward easy images which overstates model performance as we have shown. We encourage the community to develop novel dataset collection pipelines to tune datasets to desired difficulty distributions. This could take the form of posthoc filtering of webscraped images, or more on-line approaches which update collection parameters to discover ideal procedures for collecting difficult images. As shown here, there is promise in developing MVT proxies so that difficulty can be determined for each image efficiently.

Since posting this work online, MVT-based image difficulty has already been used in novel ways, for example, to improve performance by more efficiently allocating limited resources in embedded settings [40]. This highlights how useful, reproducible, and widely-applicable MVT is.

With modifications to our experiment, an MVT difficulty metric could be created for multi-object classification—for example, by evaluating class recall rather than single-class accuracy—segmentation or optical flow. Other tasks are out of scope, like visual search, as they literally require multiple saccades. Calibrating our field to what humans can do across a wide range of tasks, datasets, conditions, remains a significant challenge, but one that we think can now be addressed.

### 5.1 Acknowledgements

We would like to thank Jim DiCarlo, Guy Gaziv, Michael Lee, Dan Yamins, Colin Conwell, Martin Schrimpf and Ko Kar for their helpful feedback and discussion about our experiments. We would also like to thank David Lu for his contributions to our early experiments and directions.

This work was supported by the Center for Brains, Minds, and Machines, NSF STC award CCF-1231216, the NSF award 2124052, the MIT CSAIL Machine Learning Applications Initiative, the MIT-IBM Watson AI Lab, the DARPA Artificial Social Intelligence for Successful Teams (ASIST) program, the DARPA Knowledge Management at Scale and Speed (KMASS) program, the United States Air Force Research Laboratory and the Department of the Air Force Artificial Intelligence Accelerator under Cooperative Agreement Number FA8750-19-2-1000, the Air Force Office of Scientific Research (AFOSR) under award number FA9550-21-1-0014, and the Office of Naval Research under award number N00014-20-1-2589 and award number N00014-20-1-2643. The views and conclusions contained in this document are those of the authors and should not be interpreted as representing the official policies, either expressed or implied, of the Department of the Air Force or the U.S. Government. The U.S. Government is authorized to reproduce and distribute reprints for Government purposes notwithstanding any copyright notation herein.

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

# 6 Appendix

## 6.1 Hosting and maintenance plan

This dataset will be hosted on MIT servers in perpetuity at https://objectnet.dev/mvt/ with a backup on dropbox. Our dataset collection toolbox is hosted publicly on github at https://github.com/dmayo/MVT-difficulty. A datacard for this dataset will be available at https://objectnet.dev/mvt/datacard.

## 6.2 Object classes

Test subjects were presented with images from 50 possible object classes and asked to select which object they saw. The 50 classes were hand-picked to minimize similarity between classes that could be confusing for experiment subjects. The object classes were:
Band-Aid, T-shirt, backpack, banana, cleaver, clothes iron, coffee mug, computer mouse, digital watch, doormat, dumbbell, envelope, hair dryer, hammer, lampshade, lemon, lipstick, match, mobile phone, necklace, padlock, paintbrush, paper towel, park bench, pill bottle, pillow, plastic bag, plunger, power drill, printer, racket, ruler, safety pin, salt shaker, sandal, screw, shovel, space heater, spatula, speaker, strainer, sunglasses, teddy bear, television, umbrella, vase, wallet, waste container, water bottle, whistle

## 6.3 Image Selection

After choosing object classes, we selected images for the experiment. We used all 50 images belonging to a class in the ImageNet Validation set with no additional selection step. For ObjectNet, we collected bounding box data for the images, and then randomly selected 50 images per class such that when cropped to the bounding box, the object in the image was centered and clear.

## 6.4 Image cropping procedure

1. We draw a bounding box around the object (we use existing bounding boxes for the ImageNet validation set and collect our own bounding boxes for ObjectNet from MTurk).

2. We initialize the cropping box to be the bounding box.

3. If the cropping box does not form a square, we extend the shorter side of the rectangular cropping box to form a square. If the image is not large enough to extend the shorter side of the cropping box, we pad it with black pixels to form a square.

4. We crop using the cropping box for the image. The cropped image will be a square.

5. We resize the cropped image to be 224x224 pixels.

## 6.5 Mask generation

The masks were generated following the procedure used by [41]. Specifically, a Fourier transform was applied to each image to obtain the magnitude and phase components. Then, a random array with elements sampled uniformly from [0, 1] was added to the image phase component after which the magnitude and phase components were recombined via an inverse Fourier transform to produce the mask. Each image was paired with its particular phase-scrambled mask in the experiments.

## 6.6 Experiment Procedure and Payment

Participants both in the lab and on Mechanical Turk were presented with a document informing them of the purpose, privacy, and risks associated with the experiment and soliciting their consent to participate (see fig. 10). Participants were then instructed as to how to carry out the experiment and were shown an example video as well as the list of image classes for their review before beginning. They were informed that they would not need to memorize the classes as the classes would be shown to them after each video. Participants were also encouraged to take breaks should they feel fatigued or otherwise uncomfortable. Example instructions are shown in fig. 11

After giving consent and reading the experiment overview. participants then completed two calibration steps for to estimate the size of their monitor and their distance from the screen for us to then size

Table 1: Dataset statistics

| | |
|---|---:|
| number of responses | 200,382 |
| number of images | 4,771 |
| number of presentaiton durations | 6 |
| number of response per image | 42 |
| number of objectnet images | 2415 |
| number of imagenet images | 2356 |
| number of participants | 2647 |

the videos appropriately to 8 degrees of visual angle. First, the participants are shown an image of a credit card and are asked to use a card of their own to adjust a slider to change the size of the card on the screen to the size of their card. Since credit cards are the same size around the world, this allows us to measure the pixel-to-inches ratio of the participant's monitor. Next, the participant completes a blind-spot test [37] that allows us to estimate the distance they are sitting from their screen. Together, these two measurements are sufficient to compute the desired video eccentricity. See fig. 12 for images of the calibration steps.

The estimated hourly wage for participants on Mechanical Turk and in the lab was $10/hr and $20/hr respectively with approximately $15,000 spent in total on participant compensation.

## 6.7   In-Lab Experiment Results

To corroborate our Amazon Mechanical Turk results, we selected 200 images shown to Turk workers to conduct the same experiment in a controlled laboratory setting. 12 individuals came to participate in the experiment in which they viewed and responded to all 200 images on our 144Hz refresh rate monitor with 1ms gray-to-gray time. After conducting the experiment, 3 individuals had seen each image at each of the 4 presentation times. When compared to the MTurk results for those same 200 images, the comparison is much as we would expect. The In-Lab accuracy with shortest image duration (17ms) is less than on MTurk which can likely be contributed to the use of our new, high refresh-rate monitor in the controlled environment. It is likely that MTurk workers' personal computers differ in their graphics presentation abilities which may result in the image being visible for slightly greater than 17ms on some monitors. On the other end, the in-lab experiments reported higher accuracy at the longest image duration (10s) which is also unsurprising as the in-lab participants completed the task in a controlled environment with no distractions and are likely more inclined to take the task seriously and stay focused. The results show no significant differences in accuracy at the intermediate image durations. See fig. 4 for a side-by-side comparison between MTurk and In-Lab results.

## 6.8   Dataset statistics

We collected 42 human responses for each of 5,000 images (2,500 from ImageNet and 2,500 from ObjectNet). After reviewing response, 229 images were removed due to either being unrecognizable, mislabeled, or having been seen by the same worker twice despite safeguards in place to disallow it. Additional dataset statistics are listed in table 1.

## 6.9   Preliminary COCO MVT Results

To bolster our claims about the difficulty of current datasets, we conducted the MTurk MVT experiment on a small subset of the COCO dataset. As COCO is a more visually complex dataset than many single object classification datasets, it provides a good litmus test for how our conclusions generalize to other kinds of datasets. See 9 for the results.

### 6.9.1   Image selection

To maximize the utility of our results for both computer science and neuroscience research we selected 732 images from the Natural Scenes Dataset[42], a subset of COCO for which fMRI data was collected from human participants. We used the image crops used in the NSD experiments. These crops ensure that the image is square.

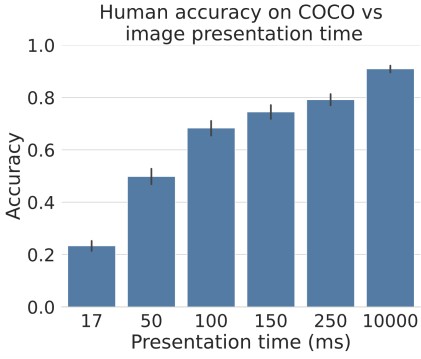
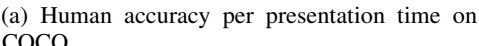
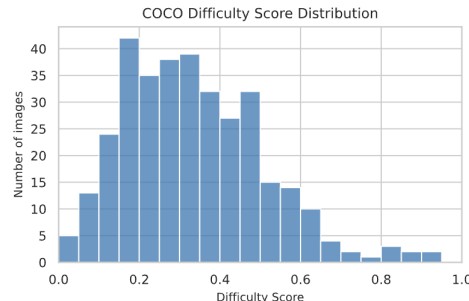

(a) Human accuracy per presentation time on COCO.

(b) Difficulty score for COCO images normalized by number of collected responses per image. A difficulty score of 1.0 here correspondes to 42 in figure 1.

Figure 9: MVT experiment results on a subset of the COCO dataset.

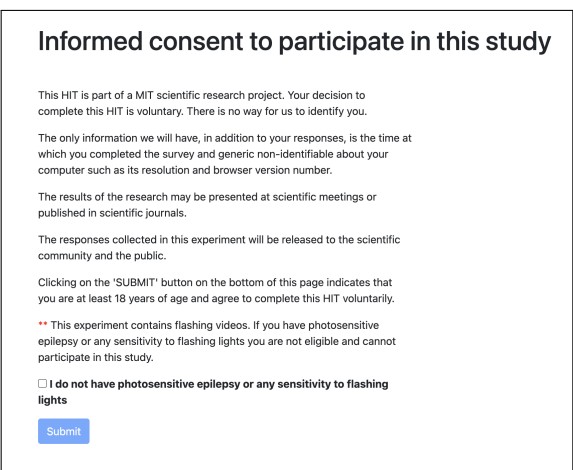

Figure 10: Informed consent page shown to participants before beginning the experiment.

### 6.9.2 Image classes

We selected a set of 41 classes such that no image contained more than one of the classes.

### 6.9.3 Experimental procedure

We conducted the MVT experiment as described in the text, asking participants to perform a 1-of-41 forced-choice single-object recognition task.

Below, we present preliminary results computed on 340 of the images. The final manuscript will include all 732 images. The results in fig. 9 are striking in their similarity to those presented for ImageNet and ObjectNet in the main text. The accuracy of human workers at each of the presentation times while performing the COCO classification task is almost the same as that of the ImageNet experiments. Given that both ImageNet and COCO originate from the same online pool of images, this is to be expected.

Similarly, the difficulty scores of COCO images (the counterpart of fig. 1) is skewed toward easy images, perhaps more so than either ImageNet or ObjectNet. These results indicate that our conclusions about the difficulty distributions of individual object recognition tasks in vision datasets generalizes. Of course, COCO has images where multiple object classes are present, which involves visual search in addition to recognizing individual image instances, but, for the quantity that we measure here, how hard are objects themselves to recognize, it COCO and ImageNet are essentially the same.

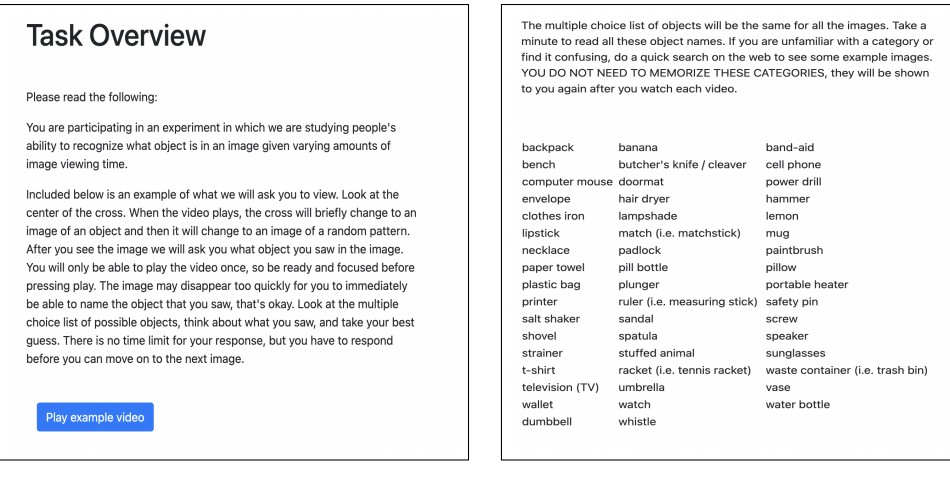

Figure 11: Instructions given to participants before beginning the experiment.

## 6.10 Finetuned Models

Here we list details regarding training/finetuning procedures for the model results reported in the paper.

### 6.10.1 Model training procedure

Pretrained models weights were instantiated using publicly available model checkpoints, either using torchvision or found on the model's source repository. The models—with the exception of CLIP—were then finetuned using subsets of the ImageNet training and validation sets containing only the 50 classes we chose to use in the psychophysics experiments. The models were finetuned for 90 epochs with an SGD optimizer and initial learning rate of 0.1 with momentum value of 0.9 and weight decay coefficient of 0.0001. The learning rate decayed by a factor of 2 every 9 epochs. Training, finetuning, and inference were performed on a cluster of 8 Nvidia TITAN RTX graphics cards.

### 6.10.2 Model Performance

We evaluate our finetuned models on the same cropped images used in our psychophysics experiments. See table 4 for model accuracy reports on the image difficulty reported in the paper and table 2 and table 3 for model performance on the full ImageNet and ObjectNet subsets of the experiment images.

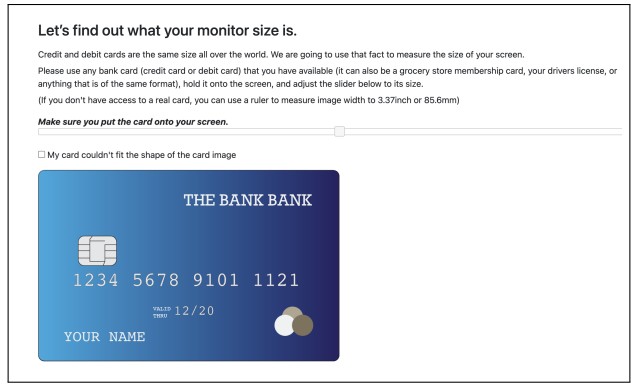

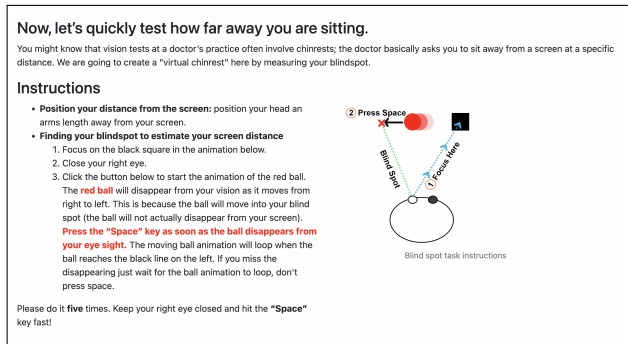

Figure 12: Images of the experiment calibration steps. The credit card task was used to measure the pixel-to-inches ratio of the subject's screen. The blind spot task provided an estimate of the subjects distance from their screen.

## 6.11 Metric calculation procedure

In this section, we go through the details in computing c-score, prediction depth, and adversarial robustness for our experiment images.

### 6.11.1 C-score

C-score [7] identifies individual image difficulty by characterizing the expected accuracy or a held-out image given training sets of varying size sampled from the data distribution. In particular, c-score is the frequency of classifying an example correctly when it is omitted from the training set. However, computing c-score for each image by brute force is computationally infeasible since we must train a separate model for each image. Instead, we computed the learning speed proxy as recommended by the authors. Learning speed measures the epoch at which an image is correctly classified by a model. Intuitively, a training example that is consistent with the training set should be learned quickly because the gradient step for all consistent examples should be similar. The authors found high Spearman rank correlation between c-score and cumulative learning speed based proxies.

We trained a ResNet-50 [43] from scratch on ImageNet1k [16] for 90 epochs with an SGD optimizer and initial learning rate of 0.1 with momentum value of 0.9 and weight decay coefficient of 0.0001. The learning rate decayed by a factor of 2 every 9 epochs and the batch size was 256. The standard ImageNet transforms were applied to all images, and the network was initialized randomly. We then evaluated our experiment images at each epoch and used the average of correct predictions as an estimated c-score for each image. fig. 13 shows the average c-scores for ImageNet and ObjectNet experiment images split by whether the ResNet-50 correctly predicted the image. C-score serves as an efficient predictor for human recognition difficulty only for images classified by the model in both ImageNet and ObjectNet. C-scores for images misclassified by the model do not reveal information about the human recognition difficulty and remain consistently low across all MVT difficulty subsets.

Table 2: Model accuracy on ImageNet per MVT subset. Models are named to include architecture, training objective, and training dataset where appropriate. ResNet-X-Y% indicates a ResNet with depth X and trained on a random Y% subset of the ImageNet-1k dataset. Model names ending in 21k were pretrained on ImageNet-21k. All other models with the exception of SWSL and CLIP models were pre-trained on the full ImageNet-1k dataset.

| MVT subset | <= 17 | <= 50 | <= 100 | <= 150 | <= 250 | <= 10000 |
|---|---|---|---|---|---|---|
| ResNet-18 | 94.4 | 91.8 | 81.1 | 77.2 | 61.3 | 49.0 |
| ResNet-18-20% | 81.9 | 77.2 | 63.7 | 58.2 | 39.8 | 34.9 |
| ResNet-18-40% | 84.4 | 85.0 | 67.8 | 63.5 | 50.5 | 46.2 |
| ResNet-18-60% | 87.5 | 88.3 | 72.6 | 70.4 | 53.8 | 46.2 |
| ResNet-18-80% | 88.1 | 86.5 | 76.7 | 68.8 | 54.8 | 48.6 |
| ResNet-50 | 94.4 | 95.5 | 85.2 | 84.7 | 79.6 | 64.7 |
| ResNet-50-20% | 86.9 | 82.9 | 72.2 | 65.1 | 48.4 | 41.0 |
| ResNet-50-40% | 93.8 | 89.1 | 74.1 | 74.1 | 60.2 | 48.2 |
| ResNet-50-60% | 91.2 | 90.2 | 78.5 | 79.4 | 60.2 | 59.4 |
| ResNet-50-80% | 90.6 | 91.5 | 83.0 | 80.4 | 68.8 | 59.0 |
| ResNet-101 | 95.0 | 95.2 | 90.0 | 87.8 | 79.6 | 71.5 |
| ResNet-101-20% | 86.2 | 85.4 | 70.0 | 66.1 | 50.5 | 48.2 |
| ResNet-101-40% | 90.6 | 90.0 | 78.1 | 77.8 | 63.4 | 50.6 |
| ResNet-101-60% | 93.1 | 89.9 | 84.4 | 77.2 | 62.4 | 59.0 |
| ResNet-101-80% | 92.5 | 94.1 | 83.0 | 82.5 | 64.5 | 61.8 |
| ResNet-152 | 93.8 | 96.4 | 93.7 | 86.8 | 78.5 | 72.7 |
| ResNet-152-20% | 86.9 | 84.4 | 73.0 | 71.4 | 52.7 | 44.2 |
| ResNet-152-40% | 93.1 | 88.4 | 76.3 | 76.7 | 62.4 | 52.6 |
| ResNet-152-60% | 93.1 | 90.4 | 82.2 | 78.8 | 66.7 | 59.4 |
| ResNet-152-80% | 90.6 | 91.9 | 86.3 | 85.7 | 76.3 | 60.6 |
| CORNet-S | 93.8 | 92.6 | 81.9 | 78.8 | 58.1 | 52.2 |
| VOneNet-Resnet50 | 93.8 | 94.4 | 84.4 | 82.5 | 67.7 | 56.6 |
| VOneNet-CORNet-S | 91.9 | 92.3 | 82.2 | 77.2 | 63.4 | 53.4 |
| VGG-19 | 91.9 | 90.2 | 80.7 | 79.4 | 62.4 | 55.4 |
| Noisy Student (EfficientNet-L2) | 95.0 | 93.3 | 87.8 | 86.8 | 68.8 | 65.5 |
| DenseNet-121 | 94.4 | 93.3 | 83.3 | 80.4 | 72.0 | 58.6 |
| MSDNet Classifier 0 | 78.8 | 76.0 | 60.0 | 54.0 | 40.9 | 33.7 |
| MSDNet Classifier 1 | 89.4 | 86.2 | 73.7 | 67.7 | 53.8 | 45.8 |
| MSDNet Classifier 2 | 91.9 | 89.9 | 77.8 | 72.5 | 62.4 | 51.4 |
| MSDNet Classifier 3 | 91.9 | 90.4 | 79.3 | 69.8 | 63.4 | 51.4 |
| MSDNet Classifier 4 | 94.4 | 91.3 | 79.3 | 78.8 | 62.4 | 52.2 |
| SimCLR ResNet50 | 88.1 | 86.3 | 73.7 | 69.8 | 60.2 | 54.6 |
| SimCLR ResNet101 | 93.1 | 89.6 | 79.3 | 83.6 | 72.0 | 57.8 |
| SimCLR ResNet152 | 93.8 | 92.1 | 83.7 | 81.0 | 72.0 | 63.9 |
| CLIP-ViT-B/32 | 95.6 | 90.8 | 79.3 | 74.6 | 67.7 | 48.6 |
| CLIP-ViT-B/16 | 97.5 | 94.7 | 83.3 | 81.0 | 80.6 | 52.2 |
| CLIP-ViT-L/14 | 98.1 | 97.1 | 92.6 | 91.0 | 86.0 | 72.3 |
| CLIP-ViT-L/14@336px | 98.1 | 96.9 | 91.5 | 92.6 | 89.2 | 73.9 |
| CLIP-ResNet-50 | 92.5 | 84.4 | 69.6 | 67.7 | 55.9 | 34.5 |
| CLIP-ResNet-101 | 94.4 | 88.1 | 71.9 | 68.8 | 67.7 | 41.0 |
| CLIP-ResNet-50x4 | 93.8 | 88.8 | 75.6 | 72.5 | 71.0 | 41.8 |
| CLIP-ResNet-50x16 | 94.4 | 91.5 | 81.1 | 77.2 | 68.8 | 43.8 |
| CLIP-ResNet-50x64 | 98.8 | 95.9 | 87.0 | 85.2 | 77.4 | 59.0 |
| EfficientNet-S | 91.2 | 92.5 | 84.1 | 78.3 | 64.5 | 62.2 |
| EfficientNet-M | 90.6 | 91.5 | 80.7 | 73.5 | 69.9 | 61.4 |
| EfficientNet-L | 95.0 | 93.0 | 87.4 | 83.6 | 75.3 | 64.3 |
| EfficientNet-S-21 | 96.9 | 95.6 | 92.6 | 87.8 | 81.7 | 71.5 |
| EfficientNet-M-21 | 97.5 | 97.0 | 93.7 | 88.9 | 84.9 | 72.3 |
| EfficientNet-L-21 | 98.1 | 96.7 | 93.0 | 90.5 | 86.0 | 73.5 |
| ViT-T/16 | 67.5 | 72.3 | 57.8 | 54.5 | 38.7 | 34.5 |
| ViT-S/16 | 95.0 | 94.5 | 82.6 | 85.2 | 68.8 | 58.6 |
| ViT-B/16 | 96.2 | 95.6 | 85.2 | 87.3 | 67.7 | 63.5 |
| ViT-L/16 | 98.8 | 97.5 | 97.4 | 96.8 | 84.9 | 80.7 |
| MoCo-V3 | 92.5 | 92.6 | 85.6 | 84.7 | 75.3 | 64.7 |
| SWSL-ResNext101-32x16d | 96.9 | 98.1 | 97.4 | 95.8 | 87.1 | 85.5 |
| SWSL-ResNet50 | 96.2 | 97.7 | 96.7 | 95.2 | 84.9 | 77.9 |
| MAE-ViT-B/16 | 94.4 | 95.6 | 88.9 | 89.9 | 77.4 | 75.1 |

### 6.11.2 Prediction depth

Prediction depth [8] represents the number of hidden layers after which the network's final prediction is already determined. The authors showed that prediction depth is larger for examples that visually appear to be more difficult and is consistent between architectures and random seeds.

We trained a linear decoder at the end of each block of a ResNet-50 on the 50 experiment classes using the ImageNet training and validation set. We used the same ResNet-50 used to calculate c-scores to ensure consistency of our results. There are 16 convolutional layers in a ResNet-50; and each linear decoder follows a convolution layer and consists of a pooling layer, flatten layer, and fully-connected layer. We use the same hyperparameters as section 6.11.1 and only updated the weights of the linear decoder.

A prediction is defined to be made at depth $L = l$ if the linear classifier after layer $L = l - 1$ is different from the network's final prediction, but the classification of the linear decoder after every

Table 3: Model accuracy on ObjectNet per MVT subset.

| MVT subset | <= 17 | <= 50 | <= 100 | <= 150 | <= 250 | <= 10000 |
|---|---|---|---|---|---|---|
| ResNet-18 | 76.1 | 65.1 | 49.1 | 41.2 | 25.3 | 20.6 |
| ResNet-18-20% | 46.2 | 44.8 | 29.6 | 27.5 | 11.5 | 12.5 |
| ResNet-18-40% | 58.1 | 53.8 | 43.0 | 35.2 | 20.7 | 16.2 |
| ResNet-18-60% | 67.5 | 60.0 | 43.3 | 37.9 | 19.5 | 17.4 |
| ResNet-18-80% | 66.7 | 63.4 | 45.7 | 37.4 | 26.4 | 17.1 |
| ResNet-50 | 80.3 | 79.7 | 62.9 | 53.3 | 44.8 | 27.0 |
| ResNet-50-20% | 58.1 | 51.1 | 36.4 | 35.2 | 24.1 | 15.7 |
| ResNet-50-40% | 70.1 | 61.7 | 45.4 | 42.9 | 29.9 | 20.3 |
| ResNet-50-60% | 70.9 | 70.1 | 54.0 | 45.1 | 33.3 | 19.7 |
| ResNet-50-80% | 76.9 | 70.4 | 53.3 | 49.5 | 31.0 | 22.0 |
| ResNet-101 | 86.3 | 81.0 | 68.7 | 54.9 | 47.1 | 29.6 |
| ResNet-101-20% | 50.4 | 53.3 | 41.2 | 33.0 | 21.8 | 13.6 |
| ResNet-101-40% | 66.7 | 61.5 | 47.4 | 35.7 | 27.6 | 18.3 |
| ResNet-101-60% | 76.9 | 70.3 | 52.9 | 46.7 | 36.8 | 22.6 |
| ResNet-101-80% | 75.2 | 75.2 | 62.9 | 46.2 | 34.5 | 25.2 |
| ResNet-152 | 85.5 | 83.8 | 68.7 | 60.4 | 46.0 | 30.7 |
| ResNet-152-20% | 58.1 | 53.7 | 39.2 | 34.6 | 19.5 | 13.6 |
| ResNet-152-40% | 66.7 | 65.1 | 49.5 | 42.9 | 25.3 | 19.1 |
| ResNet-152-60% | 72.6 | 68.4 | 57.0 | 41.8 | 35.6 | 22.6 |
| ResNet-152-80% | 74.4 | 73.5 | 55.7 | 50.0 | 35.6 | 24.6 |
| CORNet-S | 75.2 | 71.5 | 53.6 | 45.1 | 36.8 | 20.3 |
| VOneNet-Resnet50 | 77.8 | 75.7 | 59.1 | 45.6 | 35.6 | 20.9 |
| VOneNet-CORNet-S | 72.6 | 67.0 | 51.9 | 42.9 | 31.0 | 16.5 |
| VGG-19 | 76.1 | 66.3 | 50.9 | 46.7 | 34.5 | 18.3 |
| Noisy Student (EfficientNet-L2) | 76.9 | 68.7 | 54.3 | 45.1 | 26.4 | 20.9 |
| DenseNet-121 | 77.8 | 74.9 | 57.0 | 49.5 | 33.3 | 22.3 |
| MSDNet Classifier 0 | 45.3 | 39.0 | 31.3 | 28.0 | 17.2 | 10.7 |
| MSDNet Classifier 1 | 62.4 | 56.4 | 41.9 | 36.3 | 23.0 | 15.9 |
| MSDNet Classifier 2 | 70.9 | 64.3 | 52.9 | 44.0 | 28.7 | 22.3 |
| MSDNet Classifier 3 | 76.9 | 68.7 | 51.5 | 46.7 | 29.9 | 21.7 |
| MSDNet Classifier 4 | 73.5 | 70.9 | 51.5 | 47.3 | 37.9 | 22.3 |
| SimCLR ResNet50 | 60.7 | 61.0 | 49.8 | 45.6 | 27.6 | 17.4 |
| SimCLR ResNet101 | 75.2 | 70.3 | 59.5 | 55.5 | 32.2 | 25.8 |
| SimCLR ResNet152 | 78.6 | 70.3 | 57.0 | 55.5 | 35.6 | 28.7 |
| CLIP-ViT-B/32 | 88.9 | 80.5 | 61.2 | 61.0 | 43.7 | 33.6 |
| CLIP-ViT-B/16 | 92.3 | 88.2 | 78.0 | 69.8 | 50.6 | 48.4 |
| CLIP-ViT-L/14 | 97.4 | 93.8 | 88.7 | 81.3 | 78.2 | 70.1 |
| CLIP-ViT-L/14@336px | 96.6 | 94.2 | 91.1 | 85.2 | 80.5 | 70.1 |
| CLIP-ResNet-50 | 78.6 | 73.5 | 58.1 | 54.9 | 34.5 | 27.5 |
| CLIP-ResNet-101 | 86.3 | 76.9 | 63.6 | 58.2 | 41.4 | 32.5 |
| CLIP-ResNet-50x4 | 83.8 | 79.3 | 67.4 | 64.8 | 43.7 | 36.8 |
| CLIP-ResNet-50x16 | 92.3 | 85.0 | 74.2 | 69.2 | 52.9 | 48.7 |
| CLIP-ResNet-50x64 | 89.7 | 90.3 | 84.2 | 81.9 | 69.0 | 55.9 |
| EfficientNet-S | 68.4 | 66.2 | 52.9 | 40.1 | 23.0 | 20.6 |
| EfficientNet-M | 71.8 | 66.3 | 47.8 | 39.0 | 20.7 | 18.8 |
| EfficientNet-L | 75.2 | 71.5 | 54.0 | 45.1 | 31.0 | 23.2 |
| EfficientNet-S-21 | 94.0 | 83.9 | 72.9 | 64.8 | 41.4 | 30.4 |
| EfficientNet-M-21 | 88.9 | 87.5 | 71.1 | 67.0 | 46.0 | 34.2 |
| EfficientNet-L-21 | 89.7 | 86.2 | 70.1 | 68.7 | 50.6 | 36.5 |
| ViT-T/16 | 43.6 | 43.4 | 29.6 | 25.3 | 10.3 | 10.7 |
| ViT-S/16 | 83.8 | 76.8 | 57.7 | 52.2 | 31.0 | 23.5 |
| ViT-B/16 | 86.3 | 80.2 | 65.6 | 58.2 | 37.9 | 26.4 |
| ViT-L/16 | 96.6 | 92.5 | 84.9 | 81.3 | 58.6 | 44.9 |
| MoCo-V3 | 82.9 | 73.3 | 59.1 | 54.9 | 34.5 | 24.6 |
| SWSL-ResNext101-32x16d | 94.0 | 94.5 | 89.3 | 85.7 | 62.1 | 57.4 |
| SWSL-ResNet50 | 94.0 | 89.4 | 83.8 | 72.0 | 52.9 | 47.2 |
| MAE-ViT-B/16 | 84.6 | 82.7 | 69.4 | 63.2 | 37.9 | 32.5 |

layer $L \geq l$ are equal to the final classification of the network. Images classified by all decoders are said to be predicted at layer 0. Note that prediction depth is independent of whether the final prediction is correct or not. It measures the layer at which an image's prediction converges.

Figure 13 shows the average c-scores for ImageNet and ObjectNet experiment images split by whether the ResNet-50 correctly predicted the image. Like c-score, prediction depth serves as an efficient predictor for human recognition difficulty only for images classified by the model in both ImageNet and ObjectNet.

### 6.11.3 Adversarial robustness

We measured an image's distance to the decision boundary of a network using fast gradient sign method (FGSM) [9]. FGSM creates an modified example that maximizes the loss using the gradients of loss with respect to the input image:

$$mod_x = x + \epsilon \cdot \text{sign}(\nabla_x J(\theta, x, y))$$

where $adv_x$ is the modified image, $x$ is the original image, $y$ is the original input label, $\epsilon$ is a multiplier adjusted accordingly to control the size of modification step, $\theta$ is the model parameters, and $J$ is the

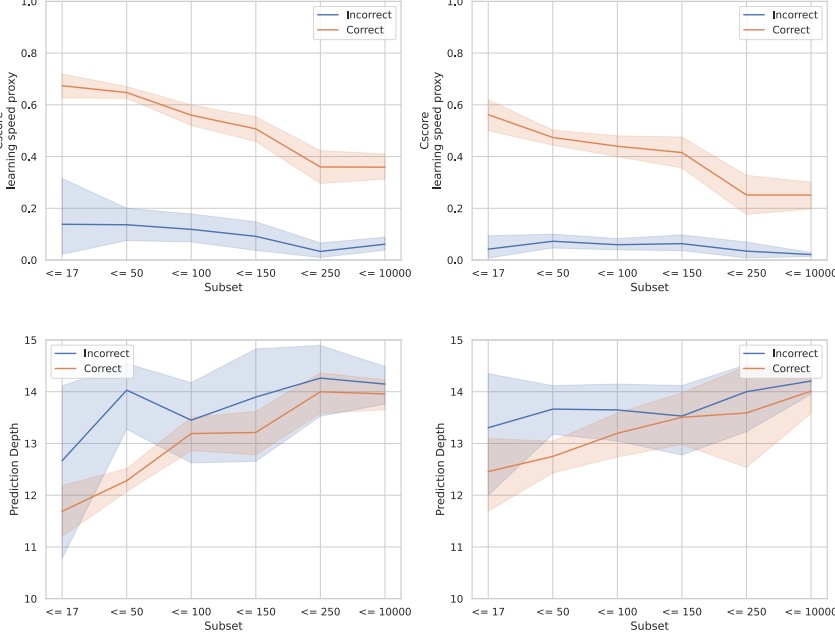

Figure 13: **Top**: left and right are average c-score over subsets for experiment ImageNet and ObjectNet images respectively. Orange shows the images that are correctly predicted by the ResNet-50 while blue shows the images that are incorrectly predicted. **Bottom**: prediction depth plots shown in the same way as top.

loss function. Note that gradients are taken with respect to the input image, and model parameters remain constant.

For an image classified by a model, we define its distance to the closest decision boundary of the model as the minimum $\epsilon$ needed for the model to misclassify the modified image. On the other hand, for an image misclassified by a model, we define its distance to the closest decision boundary of the model as the minimum $\epsilon$ needed for the model to classify the modified image.

We used the same ResNet-50 used to calculate c-scores to ensure consistency of our results. We finetuned the ResNet-50 on the 50 experiment classes using the ImageNet training and validation set. We used the same hyperparameters as section 6.11.1 and only updated the weights of the final pooling, flatten, and fully-connected layer. We used this finetuned ResNet-50 as the backbone for adversarial perturbation and correction.

While perturbing each classified image, we searched for the smallest $\epsilon$, from 0 to 0.02 incrementing by 1.25e-5 and from 0.02 to 2.5 incrementing by 0.005, that would result in a misclassification. We only applied only one gradient step when perturbing. While correcting each misclassified image, we searched for the smallest $\epsilon$, from 0 to 0.001 incrementing by 1.25e-6 and from 0.001 to 0.05 incrementing by 1.25e-5. We applied two gradient steps when correcting because correction requires finer and more steps.

Note that the search range depends on the backbone model and the dataset. One must choose them through manual trial-and-errors to yield interesting and significant results. Recall that after removing images that were incorrectly annotated, incorrectly cropped, etc section 3, we reduced to 4,771 images from the original 5,000. Of these, 3,296 and 1,475 images were classified and misclassified by the finetuned ResNet-50 respectively. We were not able to find an $\epsilon$ for every image while perturbing and correcting in the corresponding search range. We omitted these images in our analysis. We were able to successfully perturb 2,815 out of 3,296 classified images and correct 1,114 out of 1,475 misclassified images.

We hypothesized that difficult images that are classified and misclassified would be closer and further from the decision boundary respectively. fig. 8 confirms the prior hypothesis. We could not confirm

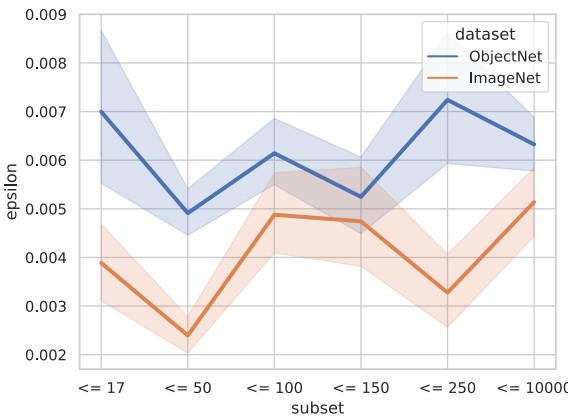

Figure 14: Average $\epsilon$ magnitude required to correct misclassified images back to their correct class per subset

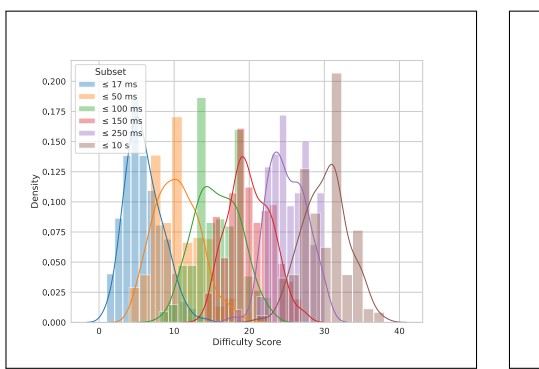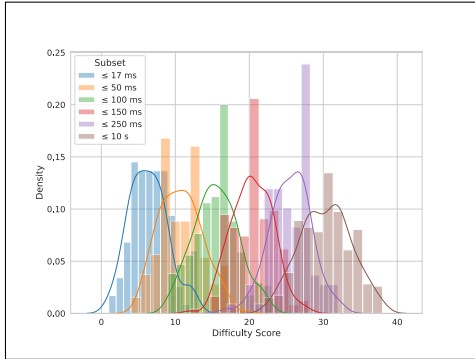

Figure 15: Distribution of difficulty score for each MVT subsets in ImageNet (left) and ObjectNet (right).

the latter hypothesis due to the smaller number of misclassified images across all subsets, as shown through the higher error bars in fig. 14

## 6.12 What factors effect MVT? Imagenet-x analysis

We found no clear trends across MVT subsets for the 16 dimensions labeled in the imagenet-x dataset. The results of our analysis can be found in table 5.

## 6.13 Constructing a metric for image difficulty

We propose two metrics:

1. Difficulty score which provides an exact ranking from most difficult to recognize to least difficult to recognize based on each response

2. six minimum viewing time (MVT) subsets that quantify the minimum amount of time required for the majority of participants to reliably recognize an image.

Difficulty score is a value from 0 to 42 that represents the number of incorrect predictions given by participants in our experiment across all timings for a particular image. Each image in our experiment was seen an equal number of times per timing and and only rarely were images that were recognizable at shorter timings also recognizable at longer timings. This results in a low difficulty score indicating that an image is easy to recognize and a high difficulty score indicating that an image is hard to recognize. These scores correlate well with the MVT difficulty subsets as shown in fig. 15. Difficulty score varies significantly by object class as well (see fig. 16).

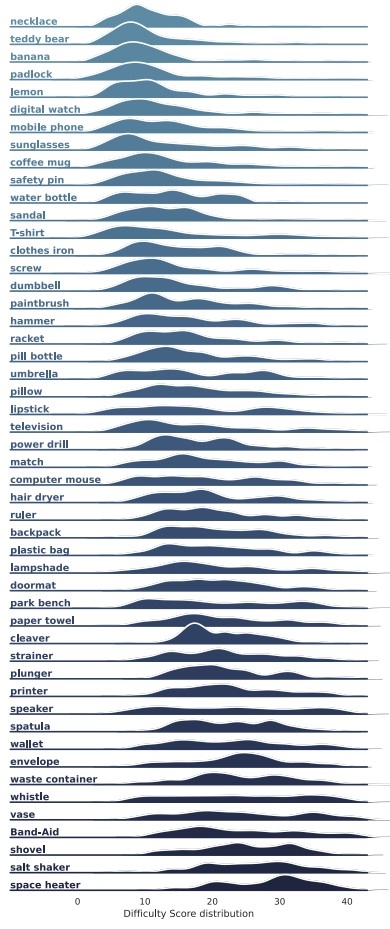

Figure 16: Difficulty distribution by object class sorted in order of increasing mean

## 6.14 Difficulty score distribution by object class

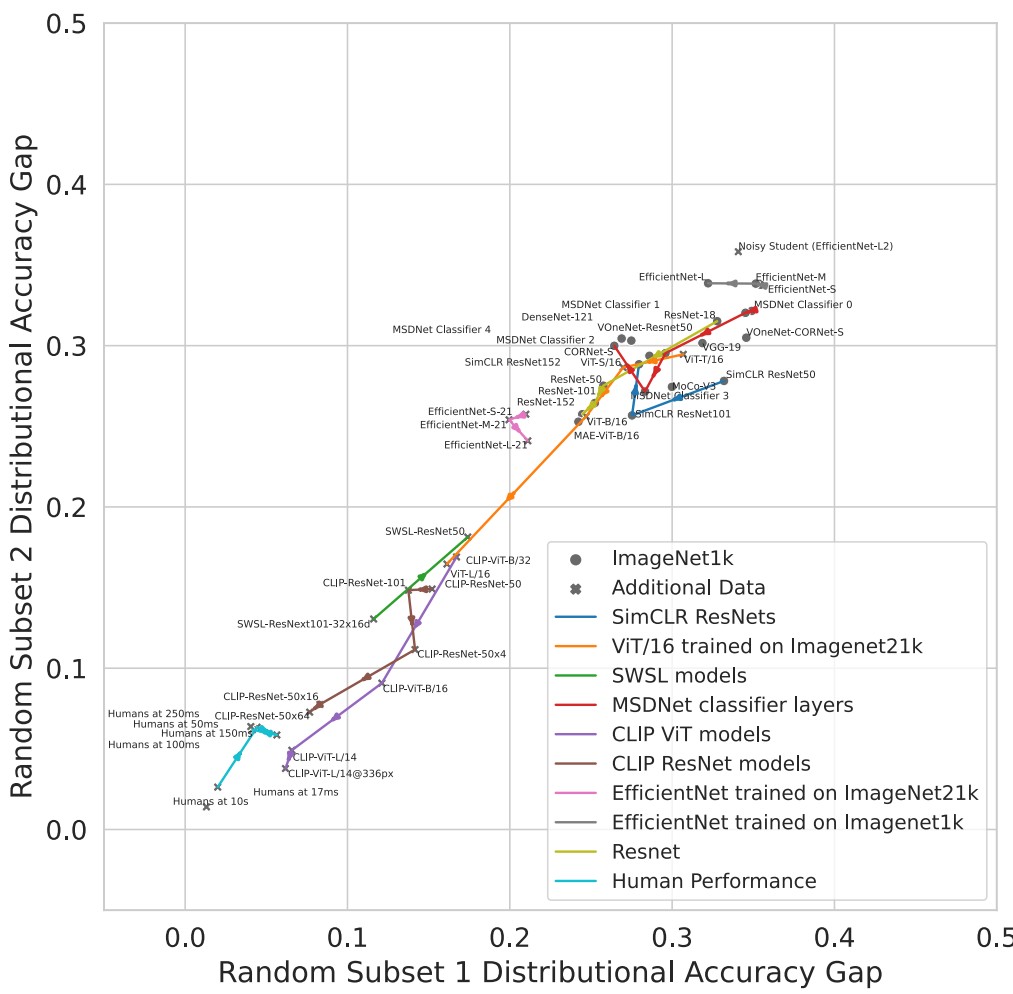

Figure 17: Robustness gap for our finetuned models on two randomly sampled subsets of our experiment data, balanced between ImageNet and ObjectNet. Lines connect model families with arrows pointing in direction of increasing model capacity. Compare with fig. 7.

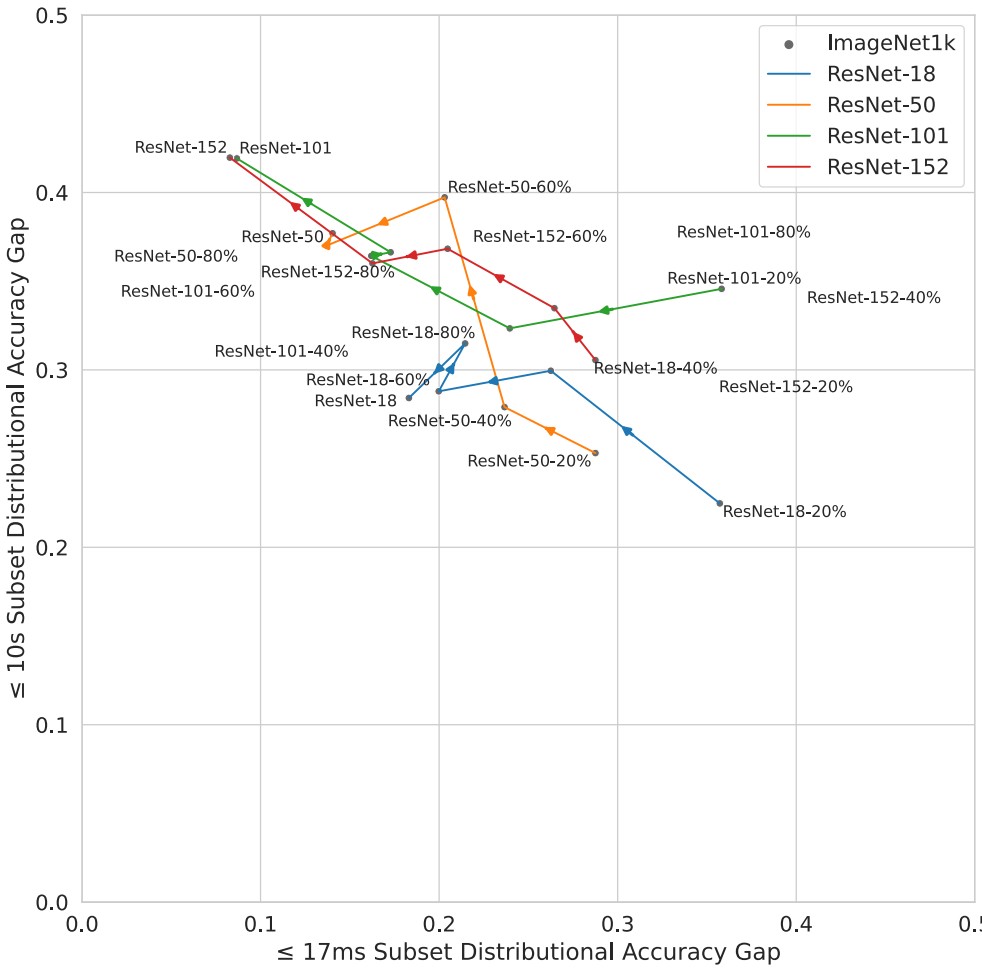

Figure 18: Robustness gap for our finetuned ResNets trained on varying percentages of the ImageNet training set. Lines connect the same architectures with arrows pointing in direction of increasing dataset percentage. Compare with fig. 7.

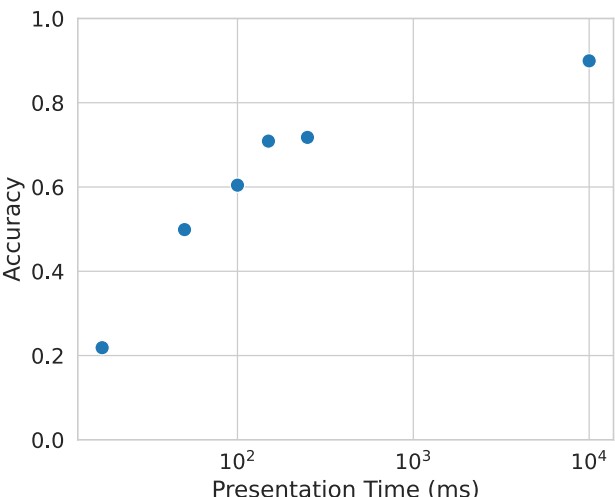

Figure 19: Human accuracy vs Image presentation time from Mechanical Turk results. Time is log-scale.

Table 4: Model accuracy on the ImageNet and ObjectNet subsets of our 4,771 images.

| | | |
|---|---|---|
| ResNet-18 | 80.4 | 48.8 |
| ResNet-18-20% | 65.0 | 31.1 |
| ResNet-18-40% | 71.5 | 39.9 |
| ResNet-18-60% | 75.3 | 43.6 |
| ResNet-18-80% | 75.6 | 45.2 |
| ResNet-50 | 87.1 | 60.0 |
| ResNet-50-20% | 71.0 | 38.3 |
| ResNet-50-40% | 77.7 | 46.4 |
| ResNet-50-60% | 80.4 | 51.6 |
| ResNet-50-80% | 82.9 | 52.6 |
| ResNet-101 | 89.2 | 62.7 |
| ResNet-101-20% | 73.2 | 37.8 |
| ResNet-101-40% | 79.5 | 45.0 |
| ResNet-101-60% | 81.9 | 52.7 |
| ResNet-101-80% | 84.6 | 56.7 |
| ResNet-152 | 90.3 | 64.5 |
| ResNet-152-20% | 73.4 | 38.5 |
| ResNet-152-40% | 78.7 | 47.9 |
| ResNet-152-60% | 82.3 | 51.4 |
| ResNet-152-80% | 84.6 | 54.8 |
| CORNet-S | 81.6 | 52.6 |
| VOneNet-Resnet50 | 84.0 | 54.8 |
| VOneNet-CORNet-S | 81.3 | 48.8 |
| VGG-19 | 81.1 | 50.2 |
| Noisy Student (EfficientNet-L2) | 86.2 | 51.2 |
| DenseNet-121 | 83.7 | 55.7 |
| MSDNet Classifier 0 | 62.8 | 29.3 |
| MSDNet Classifier 1 | 74.6 | 41.6 |
| MSDNet Classifier 2 | 78.6 | 49.6 |
| MSDNet Classifier 3 | 78.9 | 51.6 |
| MSDNet Classifier 4 | 81.0 | 52.8 |
| SimCLR ResNet50 | 76.4 | 46.1 |
| SimCLR ResNet101 | 82.1 | 55.4 |
| SimCLR ResNet152 | 84.3 | 56.0 |
| CLIP-ViT-B/32 | 80.0 | 63.3 |
| CLIP-ViT-B/16 | 84.5 | 73.6 |
| CLIP-ViT-L/14 | 92.1 | 85.8 |
| CLIP-ViT-L/14@336px | 92.0 | 86.9 |
| CLIP-ResNet-50 | 71.6 | 56.8 |
| CLIP-ResNet-101 | 75.3 | 61.4 |
| CLIP-ResNet-50x4 | 77.5 | 64.6 |
| CLIP-ResNet-50x16 | 80.1 | 72.4 |
| CLIP-ResNet-50x64 | 87.1 | 79.4 |
| EfficientNet-S | 83.1 | 48.6 |
| EfficientNet-M | 81.7 | 47.2 |
| EfficientNet-L | 85.8 | 52.7 |
| EfficientNet-S-21 | 90.2 | 66.4 |
| EfficientNet-M-21 | 91.4 | 68.2 |
| EfficientNet-L-21 | 91.5 | 68.8 |
| ViT-T/16 | 59.4 | 29.9 |
| ViT-S/16 | 84.4 | 57.1 |
| ViT-B/16 | 86.5 | 61.3 |
| ViT-L/16 | 94.1 | 78.1 |
| MoCo-V3 | 85.5 | 56.6 |
| SWSL-ResNext101-32x16d | 95.1 | 82.3 |
| SWSL-ResNet50 | 93.4 | 75.4 |
| MAE-ViT-B/16 | 89.5 | 65.2 |

Table 5: ImageNet-x factors as a % of MVT subset. Each table entry represents the percentage of the images in MVT subset (row) that were labeled as containing a feature (column). This analysis is over the ImageNet images in our dataset.

| MVT subset | multiple objects | background | color | brighter | darker | style | larger | smaller |
|---|---|---|---|---|---|---|---|---|
| 17 ms | 0.00 | 20.69 | 22.76 | 0.69 | 0.69 | 0.00 | 0.00 | 0.00 |
| 50 ms | 0.15 | 25.23 | 20.39 | 0.00 | 0.15 | 0.15 | 0.15 | 3.32 |
| 100 ms | 0.00 | 28.23 | 16.13 | 0.00 | 0.00 | 0.00 | 0.00 | 5.65 |
| 150 ms | 0.00 | 25.56 | 16.67 | 0.00 | 1.11 | 0.00 | 0.56 | 4.44 |
| 250 ms | 0.00 | 29.89 | 12.64 | 0.00 | 0.00 | 0.00 | 0.00 | 3.45 |
| 10 sec | 0.00 | 27.27 | 16.94 | 0.00 | 1.24 | 0.00 | 0.41 | 4.13 |

| MVT subset | object blocking | person blocking | partial view | pattern | pose | shape | subcategory | texture |
|---|---|---|---|---|---|---|---|---|
| 17 ms | 0.00 | 0.00 | 0.69 | 27.59 | 21.38 | 3.45 | 1.38 | 0.69 |
| 50 ms | 0.00 | 0.00 | 1.06 | 23.26 | 21.75 | 1.36 | 2.27 | 0.76 |
| 100 ms | 0.40 | 0.00 | 1.61 | 20.56 | 20.97 | 2.42 | 3.63 | 0.40 |
| 150 ms | 0.56 | 0.00 | 1.67 | 22.78 | 20.56 | 3.89 | 1.11 | 1.11 |
| 250 ms | 0.00 | 1.15 | 4.60 | 22.99 | 19.54 | 1.15 | 3.45 | 1.15 |
| 10 sec | 0.00 | 0.00 | 2.07 | 19.42 | 22.73 | 4.13 | 0.83 | 0.83 |

