# OpenReview forum: "How hard are computer vision datasets? Calibrating dataset difficulty to viewing time"
_NeurIPS.cc/2023/Track/Datasets_and_Benchmarks — NeurIPS 2023 Datasets and Benchmarks Poster_

### Official Review · Reviewer_NiF3 · 2023-06-29
**Very insightful, but potentially limited, metric for difficulty of detection datasets**

**Rating:** 7
**Confidence:** 3
**Correctness:** Yes.
**Clarity:** Yes, the paper is easy to read.

**Strengths:**

1. The experimental design, as shown in Figure 2, is efficient and logical.
2. The analysis of ImageNet and ObjectNet, which are widely used datasets, is very important and can significantly impact further research that is done with these datasets.
3. Figure 7, which depicts image difficulty vs. robustness, is very insightful and provides useful information about popular models.


**Additional Feedback:**

None.

**Documentation:**

Yes.

**Ethics:**

No.

**Limitations:**

The comparison of models to the human brain seems quite unsophisticated, and it is potentially concerning if the analysis in the paper is used to make conclusions about the similarity between the two. The crowdsource and in-lab human experiments done in the paper seem like they were properly conducted.

**Opportunities For Improvement:**

1. My biggest concern is that this metric cannot be widely applied to other datasets because of the cost/difficulty of using crowdsource workers. While the analysis in the paper is certainly useful, it does not seem to be useful as a universal metric for all object detection datasets. I would like to know if the authors agree with this point. I have used MTurk before, and I would say that it would certainly not be easy to compute this metric for a small dataset, regardless of cost. Perhaps the authors can set up an independent platform for dataset creators to compute the metric for their datasets?
2. The fact that the images were adjusted to screen size may be a point of concern, as some reviewers may have had very large or very small screens, which would have affected their response. However, it maybe argued that at a large enough sample size, these effects will have averaged out.
3. The differences between in-lab and crowdsource worker performance is a bit concerning, and not fully justified.

**Relation To Prior Work:**

Yes.

**Summary And Contributions:**

The authors point out a very relevant problem, which is that we currently do not have many, if any, metrics for the "difficulty" of an object detection task. They propose a metric called MVT, which involves measuring the time it takes for a human crowdsource worker to identify the object(s) in an image. The authors mention that they are not the first to devise such a metric, but instead the first to compute such a metric at scale for large datasets such as ImageNet and ObjectNet. In addition to computing this metric for these datasets, the authors evaluate performance of models on these datasets and relate it to the their metric.

---

> ### Author Response · Authors · 2023-08-19
> **Author Response to Review 5**
>
> Thank you for the kind review!
>
> > My biggest concern is that this metric cannot be widely applied to other datasets because of the cost/difficulty of using crowdsource workers. While the analysis in the paper is certainly useful, it does not seem to be useful as a universal metric for all object detection datasets. I would like to know if the authors agree with this point. I have used MTurk before, and I would say that it would certainly not be easy to compute this metric for a small dataset, regardless of cost. Perhaps the authors can set up an independent platform for dataset creators to compute the metric for their datasets?
>
> We are aware of the difficulties of crowdsourcing and have gone out of our way to try to alleviate this concern.
>
> We have published an open source toolkit for automatically running this experiment. It generates videos from images, and includes the server, MTurk task landing page and a notebook for data analysis. It even includes the CSV for upload to the MTurk requester site (https://github.com/dmayo/MVT-difficulty). All the user needs are the images they want to determine MVT for.
>
> With this toolkit, we make the process for collecting MVT as simple and reproducible as possible.
>
> >The fact that the images were adjusted to screen size may be a point of concern, as some reviewers may have had very large or very small screens, which would have affected their response. However, it maybe argued that at a large enough sample size, these effects will have averaged out.
>
> We agree and were extremely concerned about this problem with monitor size! It would introduce a significant source of error to our results. We carefully eliminated this problem.
>
> Thankfully, almost everyone has an item of a standard size around the world: the debit/credit card. We ask participants to hold a card against the screen and scale a credit card image to match their real world card. This allows us to precisely and accurately derive their screen size and correct the image to be of a consistent size each time. This way, we can scale the image to occupy the same degree of visual angle for all participants regardless of monitor size.
>
> >The differences between in-lab and crowdsource worker performance is a bit concerning, and not fully justified.
>
> The difference between the in-lab and MTurk experiments is accounted for on the short timing end by monitor latencies and on the long timing end by attention deficits. In the middle the higher variance of the in lab experiments accounts for the difference. While our in-lab experiments were much better controlled, practically, gathering large amounts of data in-lab is extremely challenging. This meant we could only show in-lab subjects a fraction of the data we used for the MTurk experiments. There are differences in absolute accuracy values which are explained in the text (monitor latencies [we used an expensive gaming monitor with low latency and low gray-to-gray response time while users on MTurk had normal monitors] and attention deficits [many studies have found that users on MTurk often multitask]), but these differences do not undermine our results. Actually, the in-lab results demonstrate that in more controlled circumstances, the phenomena we see with the MTurk results and model behavior would be even more pronounced which is why our MTurk experiments are a suitable substitute.
>
> > The comparison of models to the human brain seems quite unsophisticated, and it is potentially concerning if the analysis in the paper is used to make conclusions about the similarity between the two.
>
> We did not intend to use this data to make similarity judgments between brains and machines. Merely to show that some models scale better or worse on the easier or harder images. We do agree that in the future this data could be invaluable for comparing brains and models, but it would require far more analysis just as the reviewer points out.
>
> We will update the manuscript to make this clear.

---

> > ### Comment · Reviewer_NiF3 · 2023-08-29
> >
> > Thank you for your detailed response! I maintain my rating, and I would like to emphasize that I believe that this is a good paper with lots of merit.

---

### Official Review · Reviewer_L4sU · 2023-07-21
**Quantifying human recognition difficulty using minimum viewing time**

**Rating:** 7
**Confidence:** 3
**Correctness:** The claims in the paper seem to be co…
**Clarity:** Yes, the paper is very well written.

**Strengths:**

1. Proposes to bridge the gap between human recognition performance and machine recognition performance by incorporating minimum viewing time based difficulty as a criteria.
2. Demonstrate the loss of performance of the models in case of samples that are hard to recognize for humans.

**Additional Feedback:**

It would be interesting to include recognition models rooted in neurosciences such as HMax.

**Documentation:**

Yes

**Ethics:**

No significant concerns.

**Limitations:**

This work promotes greater interaction between computer vision and object recogntion psychophysics and I can't see any severe limitations. It would be interesting to identify images where difficulty is inverse correlated for humans and machines.

**Opportunities For Improvement:**

1. Recognition models considered in this study are largely feed-forward models, it would be interesting to examine the role of recurrence to better understand the phenomenon.
2. Are there any factors such as object occupancy, contrast and illumination resulting in higher MVT?

**Relation To Prior Work:**

Yes, paper adequately links prior works.

**Summary And Contributions:**

The paper provides a dataset to characterize the  object recognition difficulty of a given image. Minimum viewing time required to recognize the object in a given image is used as a proxy to measure the difficulty.  The paper also demonstrates that image samples used in popular recognition datasets such as ImageNet and ObjectNet are skewed and do not contain a lot of hard images. It is also of interest to note a significant performance decline in the performance of the models for the harder images suggesting the importance of collecting samples with recognition difficulty as a criterion.

---

> ### Author Response · Authors · 2023-08-19
> **Author Response to Review 4**
>
> > Recognition models considered in this study are largely feed-forward models, it would be interesting to examine the role of recurrence to better understand the phenomenon.
>
> This is an important observation! We already include a few such models, the CORNet family (which are recurrent).
> But, the vast majority of models in computer vision are feed-forward (although, with the caveat that one could view very deep models as unrolled versions of a smaller recurrent network). We also focus our timings on the short end to probe this regime.
>
> Our work is motivated exactly by this concern. To open up new avenues to understand the neural mechanisms behind object recognition and reconcile the biology which features recurrence heavily with the fact that recurrence does not seem to be critical at the moment.
>
> > Are there any factors such as object occupancy, contrast and illumination resulting in higher MVT?
>
> We would love to know why images are difficult. We investigated three common measures of difficulty: c-score, prediction depth, and perturbation epsilon. These actually explain more than half of the variances in difficulty in terms of these scores. Accounting for so much of the variance is non-trivial and we believe a major step toward an explanation.
>
> I think the reviewer is getting at an important point with what kinds of explanations we can expect to provide even in principle. Accounting for half the variance with metrics from other models may not be what the reviewer expected in terms of an explanation. But, it is one. And, there may be no simple explanation in terms of concepts like pose, rotation, background, etc. that are easy to explain. Much like it is hard to explain in natural language why computer vision in general is difficult---although seemingly it should be easy/natural to go from pixels to percepts---we also lack satisfying language-based explanations for what makes an image difficult. One high level qualitative trend that we can observe from our data is that images which are easy for humans are more prototypical while difficult images are atypical in innumerable different ways. We do intend to investigate this topic further in future work.
>
> > This work promotes greater interaction between computer vision and object recogntion psychophysics and I can't see any severe limitations. It would be interesting to identify images where difficulty is inverse correlated for humans and machines.
>
> We agree, that is a very interesting direction to extend this work.

---

> > ### Comment · Reviewer_L4sU · 2023-08-25
> > **Exploring the potential factors resulting in higher MVT**
> >
> > Thank you for your response. I still believe this paper would be further strengthened if there is a better characterization of the factors contributing to greater MVT. Even if the results are negative for factor primary  suspects like Pose, contrast, clutter etc, including those could benefit further analysis. Did you observe any qualitative differences in the errors made by models with recurrence and without recurrence?

---

> > > ### Comment · Reviewer_L4sU · 2023-08-29
> > > **Rating remains the same**
> > >
> > > I would like to thank the authors for submitting their updated manuscript. Although my primary concern about the factors leading to higher MVT remains I think this dataset is useful. My rating remains the same.

---

### Official Review · Reviewer_ZsVb · 2023-07-21
**Good concept and data collection, but limited analysis and wider applicability**

**Rating:** 5
**Confidence:** 4
**Clarity:** The paper is generally written clearly.

**Strengths:**

-	The paper identifies a relevant problem in the community and provides a well thought-out solution. A quantitative metric for dataset difficulty would be helpful for characterizing such benchmarks.
-	The paper details a rigorous procedure for collecting the MVT annotations from humans that avoids a range of possible design flaws.
-	Measuring the gap between ImageNet and ObjectNet performance is a good way to avoid explicit model accuracy comparisons.
- The paper is generally well-written.

**Additional Feedback:**

N/A

**Correctness:**

The claims made in the paper appear to be correct, except for a few generalizations regarding the relative difficulty of vision datasets in the field. The system for collecting MVT values is sound.

**Documentation:**

Yes, the paper clearly covers the data collection process and data is well-documented on the linked website.

**Limitations:**

Yes, the limitations section is appropriate.

**Opportunities For Improvement:**

-	It is unclear how well this approach would apply to image datasets beyond object datasets. As vision models improve, the field continues to move away from ImageNet-type datasets and towards more complex vision data such as CoCo. How could MVT be used for images that, for example, include multiple objects?
-	The paper states that “most datasets consist almost entirely of short MVT images.” How can this be verified without collecting MVT judgments on a wider range of datasets? Given the scope of these claims, It would be appropriate to evaluate, or at least consider, a wider range of image datasets beyond ImageNet and ObjectNet.
-	The paper contains a rather low number (21) of citations given the wide range of literature that considers the problem of low relative difficulty of computer vision datasets and tasks. The paper also omits relevant research in human judgment collection for image datasets.
-	Fig. 7 is difficult to interpret.
-	It isn’t clear how MVT can be easily utilized in dataset construction. The paper introduces MVT as a preventative measure for ensuring balanced dataset construction, but the judgment collection process appears to function as a fairly involved posthoc evaluation procedure.
-	It would be nice to include an analysis of the images that are deemed difficult. What makes these images difficult, and why? What is the distribution of these different factors across the dataset?


**Relation To Prior Work:**

As discussed above, the paper does not cite sufficient relevant work in adversarial image datasets, human judgments for vision data, and comparison between human and model performance in vision tasks.

**Summary And Contributions:**

The paper tackles the problem of vision datasets lacking a “real world difficulty” quantification by introducing the Minimum Viewing Time (MVT), a metric through which vision datasets can be assessed in terms of objective difficulty. The authors evaluate ImageNet and ObjectNet using this metric, and evaluate benchmark performance as a function of image MVT score.

---

> ### Author Response · Authors · 2023-08-19
> **Author Response to Review 3**
>
> > It is unclear how well this approach would apply to image datasets beyond object datasets. As vision models improve, the field continues to move away from ImageNet-type datasets and towards more complex vision data such as CoCo. How could MVT be used for images that, for example, include multiple objects?
>
> MVT can be collected for images with multiple objects by limiting the response options to include only one of the objects that is present (a particularly salient one, perhaps). We have also considered a recall task in which the worker attempts to select all the object classes they saw in the image. Other work has used visual search time (Ionescu et al.). There are a number of extensions from single-object to multi-object human difficulty experiments.
>
> We are collecting MVT for a subset of CoCo now and will update the manuscript when it is complete, but we do not believe that the results will be different from ImageNet. Particularly because ObjectNet was specifically designed to be harder than ImageNet but still has a very similar MVT distribution.
>
> We also note that in a sense our results breathe new life into ImageNet and related datasets and remind us not to declare victory too soon. They have subsets of images that are human-hard and that models have difficulty with while saturating accuracy on human-easy images.
>
> Tudor Ionescu, Radu, et al. "How hard can it be? Estimating the difficulty of visual search in an image." Proceedings of the IEEE Conference on Computer Vision and Pattern Recognition. 2016.
>
> > The paper states that “most datasets consist almost entirely of short MVT images.” How can this be verified without collecting MVT judgments on a wider range of datasets? Given the scope of these claims, It would be appropriate to evaluate, or at least consider, a wider range of image datasets beyond ImageNet and ObjectNet.
>
> Based on our results we only speculate this is the case and will adjust the language to reflect this. We do so because we see ImageNet and ObjectNet as two extremes. ImageNet represents samples from the internet (there is no reason to believe that CoCo is harder than ImageNet, since it is sampled from the same data source) while ObjectNet represents datasets that are curated carefully with bias-controls and collected directly from users.
>
> The fact that ObjectNet's MVT distribution still predominantly favors short-MVT images is a strong indication that other datasets will have similar distributions. We will explain this reasoning in the updated manuscript.
>
> > The paper contains a rather low number (21) of citations given the wide range of literature that considers the problem of low relative difficulty of computer vision datasets and tasks. The paper also omits relevant research in human judgment collection for image datasets.
>
> Thank you, we will add more context and citations. There is some work on adversarial images and on inter-annotator agreement for image datasets, that we will call out. We’ve made a list below of references we are including in the updated manuscript.
>
> > Fig. 7 is difficult to interpret.
>
> We agree that figure 7 is challenging. We have tried to make it easier to understand and will update it again. But we hope that the idea of figure 7 comes across: many model families don’t scale equally on hard vs easy images but  CLIP strikingly is different.

---

> ### Author Response · Authors · 2023-08-19
> **Author Response to Review 3 (part 2)**
>
> > It isn’t clear how MVT can be easily utilized in dataset construction. The paper introduces MVT as a preventative measure for ensuring balanced dataset construction, but the judgment collection process appears to function as a fairly involved posthoc evaluation procedure.
>
> Imagine collecting an entirely new large scale image dataset by capturing images from scratch. Without any image difficulty feedback to guide dataset collection you will likely end up with a dataset that is skewed towards containing mostly human easy images, as was the case with ImageNet and ObjectNet. With MVT measurements as a feedback signal on subsets of the collected data as it is collected, a much more difficult and balanced dataset could be created. While it is challenging to identify single factors that significantly affect image difficulty it is clear after gathering a small set of images which look very prototypical and images which deviate significantly from that class prototype. Once these prototypical images and highly atypical images have been visualized it becomes clearer how data collection strategies and procedures can be adjusted to gather more atypical looking images. While this is not a one size fits all specific tuning procedure, like MVT itself it relies on human judgment to interpret the feedback signal and adjust data collection.
>
> Another direction for future work would be incorporate the MVT dataset difficulty feedback procedure into a fully automated dataset collection system using proxies for MVT. We show that you can, to an extent, approximate MVT with metrics derived from current models. One could use this to evaluate images as they come in and tune/filter images to the desired MVT difficulty
>
> Lastly, MVT could even be used in a web scraping scenario. As you collect the dataset, you can measure its MVT. By periodically measuring MVT on subsets of data as it is collected the procedure for searching for images and deciding which images to keep during the data validation process could be adjusted.
>
> We agree that these collection procedures would be involved and that posthoc evaluation procedure over a full training set using MVT can be costly, but the more that is invested in gathering the training data and the more safely critical the data the more important evaluating MVT becomes.
>
>
> > It would be nice to include an analysis of the images that are deemed difficult. What makes these images difficult, and why? What is the distribution of these different factors across the dataset?
>
> We agree! We would love to know why images are difficult. We investigated three common measures of difficulty: c-score, prediction depth, and perturbation epsilon. These perform far above chance and account for a significant amount of the variance in difficulty. We believe this is a major step toward an explanation and is an avenue we plan to build on further in future work.
>
> I think the reviewer is getting at an important point with what kinds of explanations we can expect to provide even in principle. Accounting for half the variance with metrics from other models may not be what the reviewer expected in terms of an explanation. But, it is one. And, there may be no simple explanation in terms of concepts like pose, rotation, background, etc.  that are easy to explain. Much like it is hard to explain in natural language why computer vision in general is difficult---although seemingly it should be easy/natural to go from pixels to percepts---we also lack satisfying language-based explanations for what makes an image difficult. One high level qualitative trend that we can observe from our data is that images which are easy for humans are more prototypical while difficult images are atypical in innumerable different ways. We do intend to investigate this topic further in future work.

---

> ### Author Response · Authors · 2023-08-19
> **Author Response to Review 3 (part 3)**
>
> > As discussed above, the paper does not cite sufficient relevant work in adversarial image datasets, human judgments for vision data, and comparison between human and model performance in vision tasks.
>
> We thank the reviewer for pointing out this weakness of our work and encouraging us to engage more with previous work. We have identified some of the related work, cite it below, and will include it in the final manuscript.
>
>
>
>
> Peterson, Joshua C., et al. "Human uncertainty makes classification more robust." Proceedings of the IEEE/CVF International Conference on Computer Vision. 2019.
>
> Huber, Lukas S., Robert Geirhos, and Felix A. Wichmann. "The developmental trajectory of object recognition robustness: children are like small adults but unlike big deep neural networks." arXiv preprint arXiv:2205.10144 (2022).
>
> Geirhos, Robert, et al. "Partial success in closing the gap between human and machine vision." Advances in Neural Information Processing Systems 34 (2021): 23885-23899.
>
> Sanders, Kate, et al. "Ambiguous images with human judgments for robust visual event classification." Advances in Neural Information Processing Systems 35 (2022): 2637-2650.
>
> Jinjin, Gu, et al. "Pipal: a large-scale image quality assessment dataset for perceptual image restoration." Computer Vision–ECCV 2020: 16th European Conference, Glasgow, UK, August 23–28, 2020, Proceedings, Part XI 16. Springer International Publishing, 2020.
>
> Cui, Yin, et al. "Learning to evaluate image captioning." Proceedings of the IEEE conference on computer vision and pattern recognition. 2018.
>
> Attarian, Maria, Brett D. Roads, and Michael C. Mozer. "Transforming neural network visual representations to predict human judgments of similarity." arXiv preprint arXiv:2010.06512 (2020).
>
> Battleday, Ruairidh M., Joshua C. Peterson, and Thomas L. Griffiths. "Modeling human categorization of natural images using deep feature representations." arXiv preprint arXiv:1711.04855 (2017).
>
> Soviany, Petru, and Radu Tudor Ionescu. "Optimizing the trade-off between single-stage and two-stage deep object detectors using image difficulty prediction." 2018 20th International Symposium on Symbolic and Numeric Algorithms for Scientific Computing (SYNASC). IEEE, 2018.
>
> Tudor Ionescu, Radu, et al. "How hard can it be? Estimating the difficulty of visual search in an image." Proceedings of the IEEE Conference on Computer Vision and Pattern Recognition. 2016.
>
> Liu, Dingding, et al. "Estimating image segmentation difficulty." Machine Learning and Data Mining in Pattern Recognition: 7th International Conference, MLDM 2011, New York, NY, USA, August 30–September 3, 2011. Proceedings 7. Springer Berlin Heidelberg, 2011.
>
> Yamada, Yuki, Takahiro Kawabe, and Keiko Ihaya. "Categorization difficulty is associated with negative evaluation in the “uncanny valley” phenomenon." Japanese psychological research 55.1 (2013): 20-32.
>
> Idrissi, Badr Youbi, et al. "Imagenet-x: Understanding model mistakes with factor of variation annotations." arXiv preprint arXiv:2211.01866 (2022).
>
> Beyer, Lucas, et al. "Are we done with imagenet?." arXiv preprint arXiv:2006.07159 (2020).
>
> Yun, Sangdoo, et al. "Re-labeling imagenet: from single to multi-labels, from global to localized labels." Proceedings of the IEEE/CVF Conference on Computer Vision and Pattern Recognition. 2021.
>
> Schrimpf, Martin, Jonas Kubilius, Ha Hong, Najib J. Majaj, Rishi Rajalingham, Elias B. Issa, Kohitij Kar et al. "Brain-score: Which artificial neural network for object recognition is most brain-like?." BioRxiv (2018): 407007.

---

> > ### Comment · Reviewer_ZsVb · 2023-08-22
> >
> > Thank you for your response. Your arguments advocating for the continued exploration into traditional image classification and dataset construction are sound, and the list of additional related works also looks appropriate. I think it would be helpful to emphasize the influence that "prototypicality" has on model performance compared to that of humans within the manuscript.
> >
> > I look forward to seeing the revised submission when it is uploaded.

---

> > > ### Comment · Reviewer_ZsVb · 2023-08-30
> > >
> > > Thank you for uploading the revised manuscript. I appreciate the additions to the related works section and your discussion of prototypicality in Figure 3 and Section 4.3.
> > >
> > > I'm not able to find the MVT collection for CoCo in the revision that you proposed in the rebuttal, which would have been a strong addition to the paper as it would have demonstrated the applicability of the metric to related datasets. And, while the comparison to ImageNet-X would have been interesting if results were suggestive of some correlation, I feel that, as it is, it detracts from the strengths of the project. Given the lack of a correlation between the two, I am less confident that MVT is an appropriate metric.
> > >
> > > As my primary concerns regarding applicability and image analysis remain, I will not be changing my score.

---

> > > > ### Author Response · Authors · 2023-08-30
> > > >
> > > > > I'm not able to find the MVT collection for CoCo in the revision that you proposed in the rebuttal, which would have been a strong addition to the paper as it would have demonstrated the applicability of the metric to related datasets.
> > > >
> > > > We hope to have this done and analyzed by the discussion deadline. It will be in the final manuscript in any case.
> > > >
> > > > > And, while the comparison to ImageNet-X would have been interesting if results were suggestive of some correlation, I feel that, as it is, it detracts from the strengths of the project.
> > > >
> > > > We ask the reviewer to consider that the opposite is true.
> > > >
> > > > Had MVT merely been a function of the quantities that ImageNet-X measures, it would have been uninteresting. Just a confirmation of what many people suspect: that simple notions like pose, background, lighting, etc. determine image difficulty. It would have been a way to weigh the contributions of what is already "known".
> > > >
> > > > The fact that there is little to no correlation with ImageNet-X quantities makes MVT exciting and important!
> > > >
> > > > It means that our intuitions about what makes images hard to recognize are wrong. That image difficulty is actually a rich enterprise. That we must find new ways of explaining image difficulty that are not just in terms that we are used to like pose. This has implications for explainability (you likely cannot explain network or human failures in terms of pose/lighting/etc.), regulations (one cannot rely on someone demonstrating that their model works across object poses as a proxy for the method working in real world conditions because pose and difficulty aren't closely related), neuroscience (attempting to explain how the brain processes images in terms of these quantities is unlikely to be the best approach because that's just not what humans are sensitive to), and future work (quantifying what we are sensitive to could lead to new invariances or augmentations).
> > > >
> > > > We will add a discussion of this to the manuscript.
> > > >
> > > > > Given the lack of a correlation between the two, I am less confident that MVT is an appropriate metric.
> > > >
> > > > We don't believe the validity of MVT changes based on its correlation to ImageNet-X.
> > > >
> > > > Critically, MVT is objective. While ImagetNet-X requires many subjective judgements from annotators. For example "pose" is a particularly tricky definition in ImagetNet-X. Many other factors are subjective (such as requiring a prototypical image; which is not a well defined notion).
> > > >
> > > > Which is not to say that we're against ImagetNet-X. Our work is in the same spirit. Perhaps this could help improve the next version of ImageNet-X by determining which properties are important to annotate, which could bring MVT and ImagetNet-X into better alignment.

---

> ### Author Response · Authors · 2023-08-30
> **Preliminary COCO results uploaded!**
>
> Just letting you know that we have updated the appendix with a section outlining our preliminary results on COCO. We will include the full results and analysis in the final manuscript. Our results for COCO are very similar to those of the ImageNet experiments in the main text motivating the applicability of our method to other vision datasets.

---

### Official Review · Reviewer_cj8A · 2023-07-21
**Excellent experiment to obtain human ground truth difficulty for two image datasets**

**Rating:** 10
**Confidence:** 5
**Clarity:** The paper is very well written, no do…

**Strengths:**

- The paper is very well written, ideas flow very clearly, and concepts are presented very well.
- This paper fills an important gap in the state of the art, difficulty estimation methods are evaluated mostly qualitatively, as there is no ground truth difficulty labels, and this paper provides a human-based difficulty metric, so a new dataset of images to with human-based ground truth difficulty is built.
- The experiment for obtaining a ground truth difficulty makes sense to me, it is base on neuroscience background knowledge, and overall it seems to produce meaningful results. And it is very positive to capture data from both mechanical turk (online) and in-person lab subjects, to remove possible biases in both.
- The viewing time metric seems to be novel as applied to difficulty estimation of machine learning models, and it is used to evaluate the difficulty distribution of ImageNet and ObjectNet, with important conclusions for the literature and future research: Both ImageNet and ObjectNet are skewed in their difficulty distribution towards easy images and there is less hard images. We should aim to obtain more difficult datasets and not scale datasets with easy images.
- A major result from this paper is that ML models like ResNet and Vision Transformers seem to have lower accuracy on more difficult images, while humans do not seem to have this problem, with almost constant accuracy for varying image difficulty. The best performing model according to accuracy vs difficulty plots seem to be CLIP.
- Three difficulty estimation methods from the state of the art are compared with the ground truth difficulty obtained in the human trials, and there is a good degree of correlation, showing that there is some relation between model difficulty and human difficulty, but still new metrics and models are needed that mimic human difficulty closely.
- I believe the experimental setup, dataset building and labeling, and evaluation are correctly done and valid, and I tend to trust these results.

**Additional Feedback:**

Some questions:
- Why the large gap between stimuli intervals: 250ms to 10 s, this seems a bit odd, is there an explanation? Specially considering the other stimuly are close together with no large gaps in orders of magnitude.
- Can you describe a bit how the ImageNet/ObjectNet images were sampled, in order to make sure there is no bias in the image in terms of difficulty?


**Correctness:**

Claims are correct and supported by the data.

The dataset construction and experimental setup is well described in the paper and supplementary material, and make sense to me, and were performed correctly. I have no doubts about correctness.


**Documentation:**

This paper contains a captured dataset, and all the required information about capturing and the relevant experiments is in the supplementary. The dataset is publicly available already.

I could not find the hosting and maintenance plans in supplementary or website.


**Ethics:**

There are no ethical concerns, the experiment was supervised by an IRB.

**Limitations:**

The authors properly describe their paper' limitations in the conclusions.

**Opportunities For Improvement:**

There are not many weaknesses in this paper.
- The only weakness or unexplained concept is the gap between viewing times, from 250 ms to 10 second, the gap seems to be quite big and it is not explained, while other viewing times are much closer in time. I would expect an explanation about this.
- Only three difficulty estimation methods from the literature are used for evaluation, I believe this is a minor issue as this is not the main contribution of the paper.


**Relation To Prior Work:**

The paper makes good connections with the literature, connecting with difficulty estimation in machine learning models and neuroscience related concepts needed for experimentation. The contributions of this paper are very clear, significant, and novel.


**Summary And Contributions:**

This paper is about sample difficulty in computer vision datasets, in particular the authors perform an experimental study to obtain difficulty labels for images from human subjects, through a stimuli experiment

A subset of ImageNet and ObjectNet is used, 2500 images from each. These images are shown to lab experiment participants and online participants (Mechanical Turk), with a randomly selected stimuli ranging from milliseconds to seconds, from where the subject has to select what object class they say, which is known to correlate with image difficulty. Then a dataset is built that has ground truth difficulty labels.

The major conclusions from these experiments is that ground truth difficulty does relate to difficulty predictions made by three state of the art methods, and more difficult images have lower accuracy predicted by ML models, but not by humans, which are more robust to difficult images.

Contributions are:
- A dataset based on ImageNet and ObjectNet of 200K human judgements on difficulty based on viewing time.
- An objective and human-based difficulty metric based on viewing time.
- Estimation of difficulty distributions for ImageNet and ObjectNet.
- Experiments showing accuracy as function of the difficulty, and comparisons for biological plausibility.
- Conclusions for future research to collect more hard/difficult images, instead of just scaling the size of the datasets with easy images.

---

> ### Author Response · Authors · 2023-08-19
> **Author Response to Review 2**
>
> >There are not many weaknesses in this paper.
>
> Thank you for your review!
>
> > The only weakness or unexplained concept is the gap between viewing times, from 250 ms to 10 second, the gap seems to be quite big and it is not explained, while other viewing times are much closer in time. I would expect an explanation about this.
>
> We agree, this gap is large. It is a consequence of the significant effort required to gather such data. Each added timing requires 7 more presentations per timing for 5000 images, 35000 new presentations. It must be embedded in a larger experiment where we randomize timings to avoid a learning effect from presenting the same timing without end. We focused on the low-end because this regime is too short for eye movements and isolates fast object recognition from a more complex type of object recognition that includes saccades and visual search. This is particularly important for future neuroscience experiments which tend to focus on this fast time regime. We will include the rationale in the manuscript.
>
> >Only three difficulty estimation methods from the literature are used for evaluation, I believe this is a minor issue as this is not the main contribution of the paper.
>
> We also tried to use ImageNetX, but did not include it because we had no success. As another reviewer pointed out, we should have included this negative result. There are also only a few established difficulty measures available for us to use in these analyses.
>
> > I could not find the hosting and maintenance plans in supplementary or website.
>
> We will include the maintenance plan in an updated manuscript. The dataset will be hosted in perpetuity by our institution as well as on dropbox.
>
> >Can you describe a bit how the ImageNet/ObjectNet images were sampled, in order to make sure there is no bias in the image in terms of difficulty?
>
> We used all 50 images belonging to a class in the ImageNet Validation set with no additional selection step. For ObjectNet, we collected bounding box data for the images, and then randomly selected 50 images per class such that when cropped to the bounding box, the object in the image was centered and clear. We did minimal selection to ensure that we introduce no biases.
>
> We should have described this in the manuscript and will update accordingly.

---

> > ### Comment · Reviewer_cj8A · 2023-08-29
> > **Response to rebuttal**
> >
> > Thank you for the comments and rebuttal, this answers my questions. I just want to point out that the hosting and maintenance plan seems to be required to be in the submission, as it is part of what reviewers should take a look.

---

### Official Review · Reviewer_HVkw · 2023-07-21
**MVT metric is thoughtful, but analysis could be more robust**

**Rating:** 5
**Confidence:** 2

**Strengths:**

The proposed MVT metric is a convincing way to measure difficulty of an image. The authors seek to better understand model performance and the model human performance gap better. The number of methods and models analyzed is very thorough. The paper convincingly shows a gap in performance - that humans are still able to classify the hardest ImageNet images consistently where as models cannot. With such high accuracy on ImageNet overall, focussing on high MVT images could help measure meaningful gains in model performance.

**Additional Feedback:**

N/A

**Clarity:**

The concept of the paper is clearly explained and convincing, but portions of the paper and diagrams are challenging to follow.

The two experimental setups (MTurk and In lab) adds a level of confusion to the paper - I would advise the authors to pick one setup and include the other in the appendix. Which set of participants was uses to determine the MVT for images in Figure 6, and which set was human performance in Figure 7?

Figure 5 claims that the accuracy drops off steeply when hard images are shown for shorter time intervals. Due to the specifics of the MVT metric, this is to be expected, as if the accuracy was above 0.5 the images would not have belonged to that MVT bucket - what is the takeaway from this Figure?

Figure 6 in the main text's legends are too small to easily tell the difference between the graphs. Figure 7 of the main text is difficult to understand. It is challenging to read the name per data point, and easily digest the takeaway from the graph. The authors claim that only CLIP stands out as human like, because it has a diagonally facing line to the origin, however SimCLR does to. Both CLIP and SimCLR have scaling factors that do not have the "desirable" diagonal line.

**Correctness:**

The MVT metric is an intuitive and reasonable metric for determining the difficulty of classifying an image.

The authors very thoughtfully setup the experiment to control for as many variables as possible to ensure consistency in the difficulty of the task for users. I appreciate the thoughtfulness as to fixed distance/fixed monitor for in lab experiments, and size standardization with a credit card.



**Documentation:**

The authors provide the experimental procedure in the appendix, but do not have a formal data card for their annotations including a maintenance plan, in scope and out of scope use cases.

**Ethics:**

No.

**Limitations:**

The authors note the limitations of the Mechanical Turk experiments, especially at low time intervals. The limitation in control over the users environment and screen lead to an over-estimation of human accuracy.  The note the limitations in number of participants in the in lab experiment. The authors address the safety issues that may arise running human experiments, and state that the MIT IRB oversaw the experiment.

**Opportunities For Improvement:**

At low time intervals, the authors state that there is bias in the MTurk experiment due to the monitor refresh rate, and in lab experiment due to the number of participants. They also claim the overall accuracy of MTurk experiments is low due to annotator focus.  It is unclear which of these two situations creates a more convincingly reliable measurement.

It would be nice to include a number of samples for each MVT subset.

The paper focusses on experiments with ImageNet and ObjectNet. There are a number of related works that annotate the evaluation set of ImageNet, including ImagetNetX[1] and ImageNet REAL[2], Re-labeling ImageNet[3]. The authors could leverage these existing annotations to further the quality of their work. The authors focus on measuring difficulty, but do not investigate why images are more difficult. These works could help further the understanding and importance of MVT. [1] found that models perform worse due to changes in texture and when a person is blocking the object. For example, investigating the correlation between MVT and these factors could further expose gaps in the capabilities of models compared to humans - it could show different gaps in robustness, and could further help calibrate image difficulty.

[1] Idrissi, Badr Youbi, et al. "Imagenet-x: Understanding model mistakes with factor of variation annotations." arXiv preprint arXiv:2211.01866 (2022).

[2] Beyer, Lucas, et al. "Are we done with imagenet?." arXiv preprint arXiv:2006.07159 (2020).

[3] Yun, Sangdoo, et al. "Re-labeling imagenet: from single to multi-labels, from global to localized labels." Proceedings of the IEEE/CVF Conference on Computer Vision and Pattern Recognition. 2021.



**Relation To Prior Work:**

This work proposes a new measurement on ImageNet but does not provide sufficient use or comparison to recent work as to ImageNet evaluation.

**Summary And Contributions:**

The paper investigates vision model performance on two common benchmarks ImageNet, and ObjectNet, by judging model performance based on the difficulty of classifying an image for a human. The authors create a Minimum Viewing Time (MVT) metric based on lab and MTurk experiments, to determine how long it takes over half of the annotators to correctly classify an image. The authors compare a number SOTA models to human performance, and investigate scaling based on MVT.

The papers contributions are the MVT metric, and gathered annotations on ImageNet and ObjectNet required to compute it.

---

> ### Author Response · Authors · 2023-08-19
> **Author Response to Review 1 (part 1)**
>
> Thank you for your comments. We've responded to your concerns below.
>
> > At low time intervals, the authors state that there is bias in the MTurk experiment due to the monitor refresh rate, and in lab experiment due to the number of participants. They also claim the overall accuracy of MTurk experiments is low due to annotator focus. It is unclear which of these two situations creates a more convincingly reliable measurement.
>
> The most convincing results are the in-lab ones. They are controlled, observed, and at significant scale. Our in-lab experiments validate the MTurk experiments. The controlled in-lab environment results in the same accuracy vs presentation time trends and allows us to quantify the degree to which annotator focus and variations in at home computer screens could impact our results. Having measured this degree of difference from an ideal scenario, we believe there is merit to the MTurk experiments. Future work that builds on ours (which as we point out in the manuscript has already happened) can now do so with MTurk experiments, and adopt error bars that represent the variance between our MTurk and in-lab experiments. Each serves its own purpose: the in lab experiments demonstrate the effect, the MTurk experiments allow for engineering applications and scale.
>
> > It would be nice to include a number of samples for each MVT subset.
>
> Thank you for pointing that out. It would be a good thing to include and we will add it to the supplemental material. The MVT subset counts are:
> - 17ms: 277
> - 50ms: 1317
> - 100ms: 561
> - 150ms: 371
> - 250ms: 180
> - 10s: 594
>
> >The paper focusses on experiments with ImageNet and ObjectNet. There are a number of related works that annotate the evaluation set of ImageNet, including ImagetNetX[1] and ImageNet REAL[2], Re-labeling ImageNet[3]. The authors could leverage these existing annotations to further the quality of their work.
>
> We performed analysis with ImageNetX but we did not include results in the paper because they were negative. Pose, shape, color, etc. did not explain MVT meaningfully.  The reviewer is correct and this result is valuable; we will add this to the appendix.
>
> There are many reasons why this would be, for example, because objects are not in a sense aligned with one another (a difficult orientation for object A may be the natural simple orientation for object B) and similarly with color, shape, etc. A major new investigation with ImageNetX might uncover something, however.
>
> ImageNet REAL and Re-labeling ImageNet are almost orthogonal to our work. They each point out that the ImageNet labels have some reliability issues, which accounts for part of the variance we see in the in lab experiments (and in turn in the MTurk experiments). They would make our variances smaller, but they are already quite tight and more than clear enough for the purpose of measuring difficulty at the level we do here.
>
> We do agree that more fine-grained investigations of difficulty, potentially ones that don’t discretize viewing times, would benefit from these datasets. But those experiments quickly run into limitations of typical LCD screens.
>
> > The authors focus on measuring difficulty, but do not investigate why images are more difficult. These works could help further the understanding and importance of MVT. For example, investigating the correlation between MVT and these factors could further expose gaps in the capabilities of models compared to humans.
>
> We agree! We would love to know why images are difficult. We investigated three common measures of difficulty: c-score, prediction depth, and perturbation epsilon. These actually explain a significant amount of the variance in difficulty. Accounting for so much of the variance is non-trivial and we believe a major step toward an explanation.
>
> I think the reviewer is getting at an important point with what kinds of explanations we can expect to provide even in principle. Accounting for half the variance with metrics from other models may not be what the reviewer expected in terms of an explanation. But, it is one. And, there may be no simple explanation in terms of concepts like pose, rotation, background, etc. that are easy to explain. Much like it is hard to explain in natural language why computer vision in general is difficult---although seemingly it should be easy/natural to go from pixels to percepts---we also lack satisfying language-based explanations for what makes an image difficult. One high level qualitative trend that we can observe from our data is that images which are easy for humans are more prototypical while difficult images are atypical in innumerable different ways. We do intend to investigate this topic further in future work.

---

> ### Author Response · Authors · 2023-08-19
> **Author Response to Review 1 (part 2)**
>
> > The MVT metric is an intuitive and reasonable metric for determining the difficulty of classifying an image.
> >
> > The authors very thoughtfully setup the experiment to control for as many variables as possible to ensure consistency in the difficulty of the task for users. I appreciate the thoughtfulness as to fixed distance/fixed monitor for in lab experiments, and size standardization with a credit card.
>
> Thank you! We tried to ensure as much control as possible even in the MTurk settings.
>
> > The two experimental setups (MTurk and In lab) adds a level of confusion to the paper - I would advise the authors to pick one setup and include the other in the appendix. Which set of participants was uses to determine the MVT for images in Figure 6, and which set was human performance in Figure 7?
>
> Thank you, we should have made this clear in the manuscript and will update it. Figure 6 shows both in-lab and mturk results as separate lines. Figure 7 was created using MTurk experiment results to minimize its variance by using many more images.
>
> > Figure 5 claims that the accuracy drops off steeply when hard images are shown for shorter time intervals. Due to the specifics of the MVT metric, this is to be expected, as if the accuracy was above 0.5 the images would not have belonged to that MVT bucket - what is the takeaway from this Figure?
>
> The reviewer is right, the results in figure 5 are a necessary consequence of what MVT is.This is why we include them, they validate that our measure of MVT is accurate. We define MVT as the point at which an image becomes recognizable. If we did not see this pattern MVT would be conceptually incorrect in some way.
>
> As the reviewer points out, based on our definition of MVT, the bars below MVT will necessarily be less than 50% and the bars above MVT will necessarily be above 50%. But this does not guarantee that the plot would look the way that it does (for example, it was possible that the jump at the MVT threshold would be much less dramatic). This would be less impressive support for MVT as a human difficulty metric. But this is not the case. Instead, we see that all below MVT-threshold accuracies are very low with little accuracy increase with respect to increased viewing time and similarly, all above MVT-thresholds are high with similarly little improvements with respect to increased viewing time. Figure 5 demonstrates that MVT captures a human behavioral phenomenon that is more real than artificial which is why we consider it an important inclusion to the paper.
>
> > Figure 6 in the main text's legends are too small to easily tell the difference between the graphs. Figure 7 of the main text is difficult to understand. It is challenging to read the name per data point, and easily digest the takeaway from the graph.
>
> Good point! We will make the text more legible.
>
> >The authors claim that only CLIP stands out as human like, because it has a diagonally facing line to the origin, however SimCLR does to. Both CLIP and SimCLR have scaling factors that do not have the "desirable" diagonal line.
>
> The blue SimCLR line at the end tends upward between ResNet-101 and ResNet-152. This indicates that SimCLR doesn’t have the desirable property that CLIP does—at least most CLIP models. With only three datapoints, however, It may be that this is spurious. Although SimCLR does not clearly scale as well as CLIP, it still stands out from the other models. We will call out SimCLR as a potential model that might scale well in the updated manuscript.
>
> > This work proposes a new measurement on ImageNet but does not provide sufficient use or comparison to recent work as to ImageNet evaluation.
>
> We hope that we have resolved this concern. We actually already tried to find correlations with ImageNet X annotations as the reviewer suggested and found no meaningful correlation with image difficulty (we will add this to the manuscript). And the other ImageNet work can only lower the variance of our results, which are already sharp enough to make their point.
>
> > The authors provide the experimental procedure in the appendix, but do not have a formal data card for their annotations including a maintenance plan, in scope and out of scope use cases.
>
> This is fair. We will include a formal data card in the final manuscript. The dataset will be hosted by our institution in perpetuity with a backup on dropbox. We impose no restrictions on scope.

---

### Author Response · Authors · 2023-08-28
**Thanks for your reviews, revised manuscript coming soon**

Hello reviewers! Hope you've had a good weekend. The discussion period ends in about 30hrs so we just wanted to check in with you all.

Sorry for our delay with the revised manuscript! We know some of you are waiting to see that from us. We will be uploading that this afternoon and will notify you when we've done that.

In the meantime, we'd like to encourage you to read our responses to your reviews and let us know if you have outstanding concerns. We appreciate the effort you've put in to be communicative and engaged with us!

---

### Author Response · Authors · 2023-08-29
**Revision uploaded!**

Hello reviewers,

We've just uploaded a revised manuscript that incorporates feedback we've received in the reviews. Changes are shown in red. Please read the changes and let us know if you feel you have concerns that are yet unaddressed.

Thank you for your efforts!

---

### Decision · Program_Chairs · 2023-09-22

**Decision:**

Accept (Poster)

**Comment:**

The main contribution of this paper is a new dataset difficulty metric, MVT (Minimum Viewing Time). There were several interesting findings from this paper such as the existing image datasets generally consist of easy images, the human-level difficulty and the machine accuracy have a correlation, and most model families don't scale equally on hard/easy images (Figure 7).

After the discussion period, this paper has mixed opinions. Three out of five reviewers (cj8A, L4sU and NiF3) recommended acceptance, while two reviewers (HVkw and ZsVb) recommended borderline rejection.

The main concerns raised by the reviewers include the unreliability of the proposed metric, no investigation of the source of the difficulty, the generalizability to the other datasets rather than object datasets, and the fact that MVT judgment always requires expensive human labor.

During the reviewer-author discussion period, most concerns are resolved by the discussions. As far as I understood, the authors did provide additional materials for the concerns, but the reviewers did not respond to their last materials. When I checked the additional materials, such as the authors' rebuttal and the COCO experiments, I think the reviewers' concerns were generally well addressed.

Although some reviewers still stand for a negative decision, considering the materials provided by the authors during the rebuttal period, I recommend acceptance.